# UTILITY AS FAIR PRICING

## ABSTRACT

In 2018, researchers proposed the use of generalized entropy indices as a unified approach to quantifying algorithmic *unfairness* at both the group and individual levels. Using this metric they empirically evidenced a trade-off between the two notions of fairness. The definition of the index introduces an array of new parameters; thus, while the construction of the metric is principled, its behavior is opaque. Since its publication, the metric has been highly reproduced in the literature, researched and implemented in open source libraries by IBM, Microsoft and Amazon; thus demonstrating traction among researchers, educators and practitioners. Advice or grounded justification around appropriate parameter selection, however, remains scarce. Nevertheless, the metric has been implemented in libraries with default or hard-coded parameter settings from the original paper with little to no explanation.

In this article we take an intentionally data agnostic (rational, rather than empirical) approach to understanding the index, illuminating its behavior with respect to different error distributions and costs, and the effect of placing constraints on it. By adding the simple requirement that the the resulting fairness metric should be independent of model accuracy, we demonstrate consistency between cost sensitive learning and individual fairness in this paradigm. By viewing a classification decision as a transaction between the individual and the decision maker, and accounting for both perspectives, we prove that, with careful parameter selection, the concepts of utility and (group and individual) fairness can be firmly aligned, establishing generalized entropy indices as an efficient, regulatable parametric model of risk, and method for mitigating bias in machine learning.

## 1 INTRODUCTION

The proliferation of data driven algorithmic solutions in social domains has been a catalyst for research and development of fairness metrics in recent years. Applications in high stakes decisions in criminal justice Larson et al. (2016), predictive policing Ensign et al. (2018), healthcare Obermeyer et al. (2019), finance Mukerjee et al. (2002), employment Cohen et al. (2020) and beyond, have fueled the need for formal definition of fairness notions, metrics, and bias mitigation techniques.

Early methods for quantifying fairness, motivated by the introduction of anti-discrimination laws in the US Cleary (1968); Einhorn & Bass (1971); Cole (1973); Novick & Petersen (1976); Friedman & Nissenbaum (1996); Zliobaite (2015), fall under what has since become known as *group fairness* Barocas et al. (2019). This class of metrics considers differences in treatment (outcomes or errors) across subgroups of a population defined by *sensitive* or *protected* features. Informally, *individual fairness* is the notion that similar individuals should be treated similarly. Individual fairness is not concerned with protected features, but rather the consistency with which decisions are made Dwork et al. (2011); Zemel et al. (2013); Mukherjee et al. (2020). This tells us that for fairness, predictions must be randomized. Like these works we agree that the fairest model is the most *accurate* model, where accuracy is measured against some unknown ground truth $\tilde{Y}$, and not the target in our training data $Y$. Since the 60's the application have been

In a recent survey on fairness in machine learning, authors highlight five major dilemmas regarding progress in the space Caton & Haas (2023). The first two of these concern trade-offs between different metrics. The first discusses the difficulty in reconciling trade-offs between fairness and model performance Hajian & Domingo-Ferrer (2012); Corbett-Davies et al. (2017); Calmon et al.

(2017); Haas (2019). The second discusses trade-offs between different notions of fairness Darlington (1971); Chouldechova (2016); Kleinberg et al. (2016); Hardt et al. (2016); Murgai (2023) and the difficulty in determining which metric is most appropriate for a given problem. The latter is credited with stifling progress in the space in the 1970's Cole & Zieky (2001); Hutchinson & Mitchell (2019). Thus, clarity around the equivalence and compatibility of different fairness and performance measures are important in moving the field forward.

In 2018 Speicher et al. (2018) proposed generalized entropy indices Shorrocks (1980) as a unified measure of both *group fairness* and *individual fairness*. For any partition of a population into (mutually exclusive) subgroups, the inequality measure can be additively decomposed into a *between-group* component and a *within-group* component. The former can then be thought of a measure of *group unfairness* and the index (sum of both components), a measure of *individual unfairness*. Using the metric, they provide empirical evidence of the trade-off between group and individual fairness.

In this paper we revisit the metric proposed by Speicher et al. (2018) and mathematically prove its value in the fair measurement of risk, and regulation of it. In order to do this we use two hypothetical examples which constitute different applications of a *sociotechnical system* Barocas et al. (2019). In the first, the algorithm is *punitive*, it is used to allocate harm, by determining whether or not to incarcerate individuals on trial. In the second, the algorithm is *assistive* (or *preventative* Saleiro et al. (2019)), it is used to distribute employment opportunities. With these examples in mind, we consider the question of how an unfairness index *should* behave, knowing that a cap on the index can be efficiently integrated into any convex optimization, pre-training Heidari et al. (2018). We take an intentionally data agnostic (rational as opposed to empirical Church (2011)) approach to understanding the index. Instead we focus on the abstraction of risk, represented by generalized entropy indices, and its relationship with better known performance metrics for different index parameter choices.

The proposed index measure in the original paper increases the parametric representation of risk by the generalization parameter $\alpha$. One must define a mapping from predictions to benefits (as usual when calculating risk), and specify the generalization parameter $\alpha$. Authors in the original paper, and works that have followed, make somewhat arbitrary choices for parameters in their experiments. Thus, while the construction of the metric is principled, its behavior for different parameter choices remains opaque. Nevertheless, the metric has been implemented in open source libraries IBM (2018); Microsoft (2020); Amazon Web Services (2024). It has traction among researchers Heidari et al. (2018); Jin et al. (2023) and educators Deho et al. (2022), and is described in recent surveys Pessach & Shmueli (2022); Caton & Haas (2023). More recently Jin et al. (2023) describe a fair empirical risk minimization algorithm, in which the index is constrained during model optimization and demonstrate its promise in reducing bias.

We argue that generalised entropy indices (GEI) present a valuable family of functions (the *complete* set of subgroup decomposable functions according to Shorrocks (1980)) which warrant much closer inspection, before moving on to other welfare functions Heidari et al. (2018). We aim to prove that they parametrically extend the notion of risk, in a principled and *continuous* way that allows us to manage the multiple requirements of model accuracy, fairness (differing error costs) and between-group fairness (by choice of $\alpha$). We believe that GEI provide a parametric language ($b_{ij}$ and $\alpha$) suited to algorithmic governance at a high level. They can be computed with very little information, $(\hat{y}, y)$ or better still $(p, y)$. Such a model can be used to limit the feasible models of utility in a rational way, simply by choosing parameters reasonably and capping the index accordingly. The efficiency saving which results from using a well reasoned choice of parameters would be O($n$), since it would eliminate the need to iterate over the training data to determine the cap/threshold, which is derived analytically before training. Individual fairness, as originally defined by Dwork et al. (2011), measures similarity by the features. In order to calculate it, one must define or learn a similarity metric. This is, computationally, a significantly more expensive task Zemel et al. (2013); Lahoti et al. (2019).

The contributions of this work can be summarized as follows.

- We derive new representations of the measure, in which its relationship with important performance metrics (error rates and model accuracy and acceptance rate) are explicit. Previously these relationships were understood empirically for a limited set of parameter choices.

- We argue that in order to represent *individual fairness* as defined by Dwork et al. (2011) as faithfully as possible, the index must be orthogonal to model accuracy. For the parameter choices made Speicher et al. (2018), we show that the index is a linear function of model accuracy, and thus cannot represent individual fairness according to this constraint. We conclude that the empirical evidence presented by Speicher et al. (2018) does not support the existence of a trade-off between group and individual fairness, and more likely is a manifestation of the well documented trade-off between accuracy and fairness.

- By viewing a classification decision as a transaction between the individual and the decision maker, and accounting for the perspective of the individuals subjected to the algorithm (in addition to that of the decision maker), this work reconciles the trade-off between fairness and accuracy with a subset of utility functions (generalized entropy indices) which account for both.

- For practitioners and legislators, we provide tools to visualize the behavior of any given benefit matrix and utility function. Those readers who wish to reproduce any part of this paper, can find all relevant code and resources on GitHub. All proofs can be found in the Appendix.

The rest of this paper is organized as follows. In Section 2 we describe the metric under investigation Speicher et al. (2018); its properties, parameter requirements, calculation and decomposition. We also summarize the parameter space and datasets explored in works that have followed. In Section 3, we present analysis of several higher level representations of the index, which we use to narrow down parameter choices, to those which satisfy three specified criteria for the metric; namely that, it is independent of model accuracy, that different types of errors are appropriately weighted, and that a cap on the index corresponds to a meaningful limit on the distribution of errors. In Section 4 we discuss our findings.

## 2 MEASURING ALGORITHMIC UNFAIRNESS WITH INEQUALITY INDICES

In the standard supervised learning setting, which is is typical for high-stakes sociotechnical systems, the algorithm is learned from a data set of observations for $n$ individuals, $\mathcal{D} = \{(\boldsymbol{x}_i, y_i)\}_{i=1}^n$. For each individual $i$, we have an $m$ dimensional feature vector $\boldsymbol{x}_i \in \mathcal{X}$, a target $y_i \in \mathcal{Y}$ and a model or algorithm $\phi : \mathcal{X} \mapsto \mathcal{Y}$ which predicts the target value, given the feature vector for any individual, $\hat{y} = \phi(\boldsymbol{x})$. We shall denote the random variable $Z \in \mathcal{X}$ as the most advantaged class in our sample, indicated by those who score highest on our test. Though we focus on the case of binary classification $\mathcal{Y} = \{0, 1\}$, the work presented can be extended to consider multi-class classification $|\mathcal{Y}| > 2$ and regression $\mathcal{Y} = \mathbb{R}$ problems.

The proposed algorithmic unfairness metric is calculated for the population of $n$ individuals in two steps. First, a benefit function must be defined which maps each individual $i$ to a benefit $b_i$. Second, an inequality index $I : \mathbb{R}_{\geq 0}^n \mapsto \mathbb{R}_{\geq 0}$, is applied to the benefit array $\boldsymbol{b} = (b_1, b_2, ..., b_n)$, to measure how unequally they are distributed. The *index* provides a measure of algorithmic unfairness. The larger the value of $I(\boldsymbol{b})$, the greater the inequality. We use $\mu$ to denote the mean benefit. Below we describe each of the two steps starting with the measurement of inequality.

### 2.1 GENERALIZED ENTROPY INDICES

There are many indices $I(\boldsymbol{b})$ for measure inequality which all share the following properties:

- **Symmetry:** $I(\boldsymbol{b}) = I(\boldsymbol{b}')$ for any permutation $\boldsymbol{b}'$ of $\boldsymbol{b} = (b_1, b_2, ..., b_n)$.
- **Zero-normalization:** $I(\boldsymbol{b}) \geq 0$ and $I(\boldsymbol{b}) = 0 \Leftrightarrow b_i = \mu \ \forall \ i$.
- **Transfer principal:** Transferring benefit from rich to poor, must decrease $I(\boldsymbol{b})$, provided the individuals don't switch places in their ranking as a result of the transfer. That is, for any $1 \leq i < j \leq n$ where $b_i < b_j \ \forall \ i, j$ and $0 < \delta < (b_j - b_i)/2$, we must have $I(b_1, ..., b_i + \delta, ..., b_j - \delta, ...., b_n) < I(\boldsymbol{b})$.
- **Population invariance:** The measure depends on the distribution of benefits but not the size of the population $n$. That is, if $\boldsymbol{b}' = \langle \boldsymbol{b}, \boldsymbol{b}, ..., \boldsymbol{b} \rangle \in \mathbb{R}_{\geq 0}^{kn}$ is a $k$-replication of $\boldsymbol{b}$, then $I(\boldsymbol{b}') = I(\boldsymbol{b})$.

Generalized entropy indices are the *complete* single parameter ($\alpha$) family of inequality indices with the additional properties of subgroup decomposability and scale invariance Shorrocks (1980).

- **Subgroup decomposability:** For any partition $G$ of the population into subgroups, the measure can be additively decomposed $I(\boldsymbol{b}) = I_\beta^G(\boldsymbol{b}) + I_\omega^G(\boldsymbol{b})$ into a between-group component $I_\beta^G$, and a within-group component $I_\omega^G$. The between group component is the contribution from variations in the mean benefit, between subgroups. The within-group component is the contribution from the variation in individual benefits, within the subgroups.
- **Scale invariance:** For any constant $c > 0$, $I(c\boldsymbol{b}) = I(\boldsymbol{b})$.

**Index Calculation**   Given benefits $\boldsymbol{b} = (b_1, b_2, ..., b_n)$ with mean benefit $\mu$, the generalized entropy index can be calculated as,

$$
I(\boldsymbol{b}) = \frac{1}{n} \sum_{i=1}^n f_\alpha\left(\frac{b_i}{\mu}\right), \quad f_\alpha(x) = \begin{cases} -\ln x & \text{if} \quad \alpha = 0 \\ x \ln x & \text{if} \quad \alpha = 1 \\ \dfrac{x^\alpha - 1}{\alpha(\alpha - 1)} & \text{otherwise.} \end{cases} \tag{1}
$$

We note that the index is essentially the integral $I(\boldsymbol{b}) = \mathbb{E}[f_\alpha(B/\mu)]$, where $B$ is the random variable that generates the $b_i$ and $\mu = \mathbb{E}(B)$, computed over a discrete set of data points.

**The Generalization Parameter**   In Fig. 2, we plot the function $f_\alpha(x)$, for different choices of $\alpha$. It shows that the contribution to the index, from individuals that receive the mean benefit, is always zero, that is, $f_\alpha(1) = 0 \ \forall \ \alpha$. In addition we can show that, (i) $\alpha < 1 \Rightarrow f_\alpha'(x) < 0$, (ii) $\alpha = 1 \Rightarrow f_\alpha(x)$ is minimal at $x = e^{-1}$, (iii) $\alpha > 1 \Rightarrow f_\alpha'(x) > 0$, and (iv) $f_\alpha''(x) > 0 \ \forall \ \alpha, \ x > 0 \Rightarrow f_\alpha(x)$ is convex. Functional analysis of $f_\alpha(x)$ is presented in Appendix A.1.

The parameter $\alpha$ controls the weight applied to different parts of the benefit distribution. $f_\alpha(b_i/\mu)$ is the cost (to equality) associated with the benefit $b_i$. For $\alpha > 1$ the contribution to the index grows faster than the benefit (prioritizing equality among the rich) and slower for $\alpha < 1$ (prioritizing equality among the poor). Values of $\alpha < 1$ assert diminishing returns on benefits and thus presents a logical bound for $\alpha$ in measuring social welfare as a function of income. As $\alpha \to -\infty$, the index increasingly prioritizes the poor and the associated distribution rankings "correspond to those generated by Rawls' maximin criterion" Shorrocks (1980).

**Index Decomposition**   For any partition $G$ of the population into subgroups, the generalized entropy index $I$, is additively decomposable, into a within-group component $I_\omega^G$, and between-group component $I_\beta^G$ as follows, $I(\boldsymbol{b}) = I_\omega^G(\boldsymbol{b}) + I_\beta^G(\boldsymbol{b})$, where,

$$
I_\omega^G(\boldsymbol{b}) = \sum_{g=1}^{|G|} \frac{n_g}{n} \left(\frac{\mu_g}{\mu}\right)^\alpha I(\boldsymbol{b}_g), \quad I_\beta^G(\boldsymbol{b}) = \sum_{g=1}^{|G|} \frac{n_g}{n} f_\alpha\left(\frac{\mu_g}{\mu}\right). \tag{2}
$$

**Relative importance of between-group and within-group fairness**   From Eq. (2) we can see that the between-group component of the index $I_\beta^G$ is always a true weighted average over the subgroups, since the coefficients $(n_g/n)$ always sum to unity; however the same cannot be said for the coefficients in the within-group component, $(n_g/n)(\mu_g/\mu)^\alpha$. $I_\omega^G$, is a true weighted sum of the index values for the subgroups, only when $\alpha = 0$ or $\alpha = 1$. When $\alpha = 0$, the index value for each subgroup in the within-group component $I_\omega^G$, is weighted by the proportion of the population in the subgroup. When $\alpha = 1$, the index for each subgroup in the within-group component $I_\omega^G$, is weighted by the proportion of the total benefit in the subgroup, effectively placing proportionally greater weight on equality within wealthier groups. For $\alpha \notin \{0, 1\}$, the sum of coefficients of the within group component, is linearly dependent on the between-group component. That is, $\sum_{g=1}^{|G|} \frac{n_g}{n} \left(\frac{\mu_g}{\mu}\right)^\alpha = 1 + \alpha(\alpha - 1) I_\beta^G(\boldsymbol{b}; \alpha)$. For $\alpha \in (0, 1)$ the sum of coefficients of $I_\omega^G$ is less than unity, and minimized when $\alpha = 1/2$. Consequently, the relative contribution to the index from the between-group component is maximized when $\alpha = 1/2$. Thus between-group fairness is maximally prioritized by the index, when $\alpha = 1/2$. Here, the sum of coefficients in the within group component of the index is $1 - I_\beta^G/4$.

## 2.2 Mapping Predictions to Benefits

A key component of the measure, is the definition of the mapping from algorithmic prediction to benefit. Benefits are floored at zero and the mean benefit must be greater than zero. Benefits are

relative, they must be defined on a *ratio scale*, as oppose to an *interval scale*, to ensure that relative comparisons of benefits are meaningful. On a ratio scale, zero represents a true minimum. On an interval scale, zero is arbitrarily chosen, nevertheless differences can be interpreted meaningfully. An example is temperature, for which Kelvin is a ratio scale; Celsius and Fahrenheit are different local interval scales. If we are interested in global solutions, we should use Kelvin.

One can imagine that there is *almost* always some benefit or cost to any decision; that benefit is the information gained from the process, which guides both the benefit provider and recipient to their next decision. Every decision is useful, even the bad ones, assuming we live to learn a lesson from it. So as long as the decision is not death, and some information was shared by both parties, we can assume there is some, potentially small, positive benefit, regardless of our position. The same algorithm with a higher minimum benefit would be preferred by any reasonable measure of utility. How does one increase the minimum benefit? With more *relevant* information exchange and a path for recourse if necessary.

Given two arrays, the *target data* $\boldsymbol{y}$ and model prediction $\hat{\boldsymbol{y}}$, of size $n$, all $n$ individuals can be categorized in a confusion matrix. A benefit function can then be defined by simply assigning a non-negative benefit value, to each element of the matrix $b_{ij} = \text{benefit}(\hat{y} = i, y = j)$. Since the generalized entropy index is scale invariant, we can choose any one of these to be unity, leaving $|\mathcal{Y}|^2 - 1$ degrees of freedom in the definition of the benefit matrix.

It's easiest to reason about the matrix from the perspective of one *stakeholder* at a time. We shall assume stakeholders include three broad parties. These are, the *benefit providers*, *benefit recipients* and the *regulator*. The *decision maker* and *subject* could be the either the recipient or provider of benefits. Neither benefit provider nor recipient can see beyond the decision, under one of the two outcomes. For the employer, the cost is the same regardless of whether the chosen candidate was worthy. Similarly, the cost of incarcerating a person is the same, regardless of how much the defendant earned when they were free. From any one perspective, two of the four outcomes look the same Elkan (2001). Thus, we can reduce the complexity of the analysis, by assuming that two of the four possible outcomes $\hat{y}, y \in \{0, 1\}$ are of unit benefit. More specifically, we will assume a ternary model of benefits, where the benefit associated with an outcome could be one of three values, $b_{ij} \in \{b_-, b_+, 1\}$ and $b_{min} = b_- < b_+$. One final constraint is that of *convexity*, for which the benefit must be monotonic in $\hat{y}$ Heidari et al. (2018).

In this paper, we shall play the role of regulator. The decision maker exerts power and influence through deployment of their model at scale. They are, in some sense, the navigators and the stakeholders are (in most cases involuntary) passengers. As regulator, we must consider both perspectives. We accept the decision makers right to navigate (optimize), within reason or *risk appetite*. We must take, longer term view to protect *everyone* (including foreseeable future stakeholders) and avert disaster by constraining the direction of travel. The regulator must decide the relative importance of precision $\mathbb{P}(Y = 1|\hat{Y} = 1)$ versus recall $\mathbb{P}(\hat{Y} = 1|Y = 1)$ based on the *mission*, *context* and *law*. We can assume an unregulated decision maker would almost certainly be greedy. As the regulator, we can impose the minimum legal benefit. In some sense, every decision can be viewed as a *transaction* or *bet*; an investment (or divestment) in an *entity*, which may yield a return, (or prevent a loss). The model score is an indication of the *present value* of the subject, based on incomplete and potentially erroneous information about them. As a regulator we can preclude predatory pricing models, based on our own definition of utility, ultimately setting risk appropriate bounds on the decision space for a given application.

Table 1 summarizes the datasets and parameters explored empirically by researchers in previous works. It shows that $b_{ij} \in \{b_-, b_+, 1\}$ is sufficiently broad to encompass parameter choices made in all known prior works, summarized in Table 1 above, and more. All prior works in Table 1 assume that accurate predictions are equally beneficial, that is, $b_{ij} = \text{benefit}(\hat{y} = i, y = j) = ((1, b_-), (b_+, 1))$ where $b_\pm$ are the false positive and negative benefit respectively. Two of the papers use $b_{FN} = 0$. The choice of a zero benefit (unlike when calculating empirical risk Elkan (2001)) is problematic. From a practical perspective, it limits one's ability to set two differing relative weights; one between the error types, and another between an error verses an accurate prediction. The case where the decision has the most impact $Y = 1$, is rationally prioritized by all stakeholders. Zero benefits also prohibit choices of $\alpha \leq 0$, and it's is not clear, at this stage, why such a choice should be unreasonable.

Table 1: Explored parameter space and datasets in prior works.

| Ref | $(b_{FN}, b_{FP})$ | $\alpha$ | Adult[a] | COMPAS[b] | C & C[c] |
|---|---|---|:---:|:---:|:---:|
| Speicher et al. (2018) | $(0, 2)$ | $2$ | • | • | |
| Heidari et al. (2018) | $\left(0, \left\{\frac{3}{2}, 2\right\}\right)$ | $\left\{\frac{1}{10}, \frac{1}{5}, \frac{2}{5}, \frac{3}{5}, \frac{4}{5}, 1\right\}$ | | • | • |
| Jin et al. (2023)[d] | $\left(b_\pm = 1 \pm \left\{\frac{5}{8}, \frac{5}{9}, \frac{1}{2}\right\}\right)$ | $\{0, 1, 2\}$ | • | • | |

[a] Identifying high earners Becker & Kohavi (1996). [b] Predicting recidivism risk Larson (2016). [c] Predicting crime rates Redmond (2009). [d] Jin et al. also looked at two more datasets, predicting bar exam success Wightman (1998) and identifying individuals with prestigious occupations Van der Laan (2017).

We shall describe the proportion of individuals receiving the unit benefit as the *unit reward rate* and denote it as $\lambda$. We do not know what the rewards $b_\pm$ are, they may be more or less than unity. The unit rewards could correspond to a column, row or diagonal. In each case, $b_\pm$ correspond to different elements of the benefit matrix. In Theorem 3.3, we will see a representation of the index in terms of $\mu$ and $\lambda$ which allows us to consider any of the possibilities.

## 3 METRIC ANALYSIS

In this section we present several higher level representations of the index. The first is as a function of the mean benefit $\mu$ and the unit reward rate $\lambda$. We then consider three cases. In the first we assume that only accurate predictions yield the unit reward $\lambda = \mathbb{P}(\hat{Y} = Y)$ (as in all previous works summarized in Table 1). In the next two examples, we assume unit rewards corresponds to a row in which case the unit benefit is the maximum benefit. If the algorithm is punitive then $\lambda = \mathbb{P}(\hat{Y} = 0)$. For assistive or preventative algorithms $\lambda = \mathbb{P}(\hat{Y} = 1)$. Under each of the latter two cases, we derive the benefit function and subset of generalization parameter values for which a cap on the index, corresponds to a meaningful limit on the distribution of errors. We begin by clarifying the connection between risk and fairness.

### 3.1 RISK AND FAIRNESS

In the standard supervised learning setting, the predictions $\hat{y} = \phi(\boldsymbol{x})$ are generated by a model $\phi : \mathcal{X} \mapsto \mathcal{Y}$ which is learned via empirical risk minimization. For binary classification, we start with a model hypothesis, usually in the form of a class of parametric functions $\theta \in \Theta$, where $\theta : \mathcal{X} \mapsto (0, 1)$ maps features $\boldsymbol{x}$ to a probability $\theta(\boldsymbol{x}) = \mathbb{P}(Y = 1|\boldsymbol{x}, \theta) = p$. If we know the target value $y$, we can calculate the loss $\mathcal{L}(\theta(\boldsymbol{x}), y) \in \mathbb{R}_{\geq 0}$ for a given model $\theta$. The optimal model $\theta^*$ is that which minimizes the empirical risk (expected loss over all individuals $i$ in the training data set),

$$\theta^* = \underset{\theta \in \Theta}{\operatorname{argmin}} \left\{ \mathbb{E}_{(\boldsymbol{X}, Y)}[\mathcal{L}(\theta(\boldsymbol{X}), Y)] \right\} \quad \text{where} \quad \mathbb{E}_{(\boldsymbol{X}, Y)}[\mathcal{L}(\theta(\boldsymbol{X}), Y)] = \frac{1}{n} \sum_{i=1}^{n} \mathcal{L}(\theta(\boldsymbol{x}_i), y_i).$$

A common choice of loss function is the log loss, $\mathcal{L}_0(p, y) = -\ln \mathbb{P}(Y = y|\boldsymbol{x}, \theta) = -y \ln p - (1 - y) \ln(1 - p)$, which is defined on $p \in (0, 1]$. Another valid choice is the squared error loss, $\mathcal{L}_2(p, y) = (p - y)^2 = y(1 - p)^2 + (1 - y)p^2$, which is defined on $p \in [0, 1]$.

There is an important difference between the calculation of the index and empirical risk. To calculate the index, we divide by the mean benefit before calculating the expected value. The index can be written as $\mathbb{E}[f_\alpha(B/\mu)]$ where $\mu = \mathbb{E}[B]$ and $f_\alpha(b/\mu)$ is the contribution to the cost from an individual with benefit $b$. We can write the index $\mathbb{E}[f_\alpha(B/\mu)]$ as a function of $\mathbb{E}[f_\alpha(B)]$ as follows.

**Theorem 3.1** (Influence of the Mean Benefit on the Generalized Entropy Index)**.**

$$I(\boldsymbol{b}; \alpha) = \mathbb{E}\left[ f_\alpha\left(\frac{B}{\mu}\right) \right] = \frac{\mathbb{E}[f_\alpha(B)] - f_\alpha(\mu)}{\mu^\alpha}, \tag{3}$$

*where $B$ is the random variable that generates $b_i$ and $f_\alpha$ is defined in Eq. (1). Proof in Appendix A.2.*

Consider the simplest case, where only accurate predictions are rewarded, that is, $b(p, y) = \mathbb{P}(Y = y | \boldsymbol{x}, \theta) = yp + (1 - y)(1 - p)$. In this case, the mean benefit $\mu$ is exactly the model accuracy, and substituting Eq. (1) into (3) we see that for $\alpha = 0$, we can write the index as a function of the cross entropy loss. $I(\boldsymbol{b}; 0) = \mathbb{E}[\mathcal{L}_0(p, y)] + \ln \mu$. If we rewrite the benefit in terms of the cost, $b(p, y) = 1 - c(p, y)$, where $c(p, y) = \mathbb{P}(Y \neq y | \boldsymbol{x}, \theta) = y(1-p) + (1-y)p$. we see that for $\alpha = 2$ the index can be written as a function of the squared error loss, $I(\boldsymbol{b}; 2) = (\mathbb{E}[\mathcal{L}_2(p, y)] - (1-\mu)^2)/(2\mu^2)$. The values $\alpha = 0$ and $\alpha = 2$ represent the only two special cases of generalized entropy for which the loss $\mathbb{E}(B) = \mathcal{L}_\alpha(p, y)$ is Fisher consistent Cox & Hinkley (1974); Buja et al. (2005). That is to say that the expected loss is minimized when $\mathbb{E}(p) = \mathbb{E}(y)$, ensuring that the resulting predictor provides a statistically unbiased estimate of $Y$.

Together with Theorem 3.1 we conclude that the unfairness index is able to express linear functions of empirical risk, where the gradient and intercept depend on an additional (new) parameter $\mu$. Different values of $\alpha$ correspond to different choices of loss function. For values of $\alpha \notin \{0, 2\}$, the estimator which results from maximizing for accuracy is biased. Finally, note that when $\mu = 1$, the index is equivalent to $I(\boldsymbol{b}; \alpha) = \mathbb{E}[f_\alpha(B)] = \mathbb{E}[\mathcal{L}_\alpha(p, y)]$.

## 3.2 REPRESENTATIONS

**Theorem 3.2** (Index as a function of $\mu$ and $\lambda$). *For benefits $b_i \in \{b_-, b_+, 1\}$, the index $I(\boldsymbol{b}; \alpha)$ can be written as a function of the mean benefit $\mu$ and unit reward rate $\lambda$ as follows,*

$$I(\boldsymbol{b}; \alpha) = [(A_\alpha + \beta_\alpha)(1 - \lambda) + \beta_\alpha(\mu - 1) - f_\alpha(\mu)]/\mu^\alpha \qquad (4)$$

$$\text{where} \qquad A_\alpha = f_\alpha(b_\pm) - b_\pm \beta_\alpha \quad \text{and} \quad \beta_\alpha = \frac{f_\alpha(b_+) - f_\alpha(b_-)}{b_+ - b_-}. \qquad (5)$$

*$A_\alpha$ and $\beta_\alpha$ are respectively the intercept and the gradient of the secant line passing through $f_\alpha(b_-)$ and $f_\alpha(b_+)$. Proof in Appendix C.2.*

From Eq. (4) we see that $I(\boldsymbol{b}; \alpha)$ must depend on $\lambda$. Thus Theorem 3.2 tells us that, all benefit functions of the form $b_{ij} = ((1, b_{FN}), (b_{FP}, 1))$ will result in a metric that is dependent on model accuracy $\mathbb{P}(\hat{Y} = Y) = \gamma$. In fact, the unfairness index is proportional to the model error $1 - \gamma$ when $\mu = 1$. This result is consistent with literature which demonstrates a trade-off between model accuracy $\gamma$ and fairness Hajian & Domingo-Ferrer (2012); Corbett-Davies et al. (2017); Calmon et al. (2017); Haas (2019). Ideally, we want the unfairness index to be orthogonal to $\gamma$. Below we analyze the behavior of the metric with respect to $\mu$ and $\lambda$.

**Domain** The unit reward rate is bounded, $\lambda \in (0, 1)$. Since $b_- < b_+$ the total number of benefits is minimized when all benefits are $b_-$, and maximized when all benefits are either unity or $b_+$, depending on which is larger. Thus, for $b_- < b_+$ and unit reward rate $\lambda$, the mean benefit $\mu$ must satisfy the following bounds,

$$b_- < b_- + (1 - b_-)\lambda \leq \mu \leq b_+ + (1 - b_+)\lambda < \max(b_+, 1). \qquad (6)$$

As the unit reward rate $\lambda$ increases, the range of possible values $\mu$ can take, decreases. The domain space is then a triangle, the illustration in Fig. 1 assumes that $b_- < b_+$. If $p = \mathbb{P}(Y = 1)$ is known, the domain space is reduced to a quadrilateral with two parallel sides.

**Corollary 3.2.1** (Behavior with respect the unit reward rate). *For benefits $b_i \in \{b_-, b_+, 1\}$ and fixed mean benefit $\mu$, the index is a linear function of the unit reward rate, for $b_+ < 1$, it is increasing and for $b_+ > 1$, it is decreasing. When either of $b_\pm = 1$, the index is independent of unit reward rate. Proof in Appendix C.2.*

**Corollary 3.2.2** (Behavior with respect to the mean benefit). *For benefits $b_i \in \{b_-, b_+, 1\}$ and fixed $\lambda$, the index has a single turning point at $\mu = \hat{\mu}(\lambda)$, where,*

$$\hat{\mu}(\lambda) = \begin{cases} -1/\beta_0 & \text{for} \quad \alpha = 0 \\ (A_1 + \beta_1)\lambda - A_1 & \text{for} \quad \alpha = 1 \\ \dfrac{\alpha(\alpha - 1)[(A_\alpha + \beta_\alpha)\lambda - A_\alpha] - 1}{(\alpha - 1)^2 \beta_\alpha} & \text{otherwise.} \end{cases} \qquad (7)$$

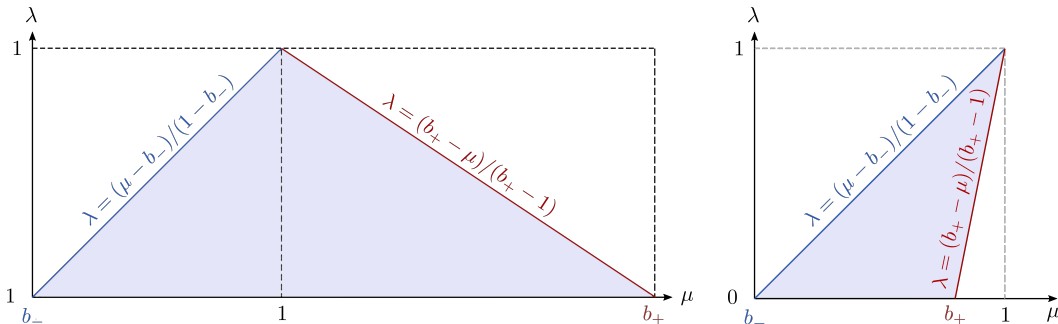

Figure 1: Visualization of index domain assuming $b_+ > 1$ and $b_+ < 1$ respectively.

*In most cases the turning point is a maximum turning point. In the special case where $b_- = 0$, as we increase $b_+$, the turning point changes from a minima (for $b_+ < 1$) to an inflection point (at $b_+ = 1$), and finally a maxima (for $b_+ > 1$). Proof in Appendix C.2.*

Corollary 3.2.1 and 3.2.2, show that the unfairness index can exhibit a wide enough variety of behaviors that, poorly chosen parameters could result in a metric that behaves nonsensically. Next we derive expressions for the index in terms of the error rates under different interpretations of the unit rate $\lambda$.

**Theorem 3.3** (Index as a function of the error rates for $\lambda = \mathbb{P}(\hat{Y} = Y)$). *For the benefit function $b_{ij} = ((1, b_{FN}), (b_{FP}, 1))$, where $0 \leq b_{FN} < 1 < b_{FP}$, the index $I(\boldsymbol{b}; \alpha)$ can be written as a function of the false positive (FPR) and false negative (FNR) rates,*

$$I(\boldsymbol{b}; \alpha) = [f_\alpha(b_{FN})pFNR + f_\alpha(b_{FP})qFPR - f_\alpha(\mu)]/\mu^\alpha \tag{8}$$

$$where \qquad \mu = 1 + (b_{FN} - 1)pFNR + (b_{FP} - 1)qFPR \tag{9}$$

$p = \mathbb{P}(Y = 1)$ and $q = p - 1$. *Proof in Appendix C.2.*

Theorem 3.3 shows that the data reward rate $p = \mathbb{P}(Y = 1)$ affects the relative weight of $FPR$ and $FNR$ in $\mu$, and $f_\alpha(b_{F\pm})$ in $I(\boldsymbol{b}; \alpha)$.

Next we consider the two cases where the unit rewards correspond to a row, making the mean benefit orthogonal to model accuracy. For convexity both $b_\pm$ must be less than unity. Intuitively, we know that if $b_-$ is close to zero and $b_+$ is close to one, whichever a benefit of $b_-$ will have the greatest cost to equality, and so this is best placed on the error type we wish to avoid in the confusion matrix. For punitive algorithms, we wish to avoid false positives, and for assistive

**3.2.1 Avoiding harm with punitive algorithms**  In this example the decision maker incarcerates high risk subjects. Thus, benefits should be decreasing in $\hat{y}$. As regulator, we wish to avoid false positives, thus $\lambda = \hat{q} = \mathbb{P}(\hat{Y} = 0)$ and $(b_-, b_+) = (b_{FP}, b_{TP})$. From Eq. (4) we know that,

$$I(\boldsymbol{b}; \alpha) = [(A_\alpha + \beta_\alpha)\hat{p} + \beta_\alpha(\mu - 1) - f_\alpha(\mu)]/\mu^\alpha. \tag{10}$$

**Theorem 3.4** (Index as a function of the error distribution for $\lambda = \hat{q} = \mathbb{P}(\hat{Y} = 0)$). *For the benefit function $b_{ij} = ((1, 1), (b_{FP}, b_{TP}))$, where $b_{FP} < b_{TP} \in (0, 1)$, the index $I(\boldsymbol{b}; \alpha)$ can be written as a function of the false negative (FNR) and positive (FPR) rates,*

$$I(\boldsymbol{b}; \alpha) = [p(1 - FNR)f_\alpha(b_{TP}) + qFPRf_\alpha(b_{FP}) - f_\alpha(\mu)]/\mu^\alpha \tag{11}$$

$$\mu = 1 - (1 - b_{TP})p(1 - FNR) - (1 - b_{FP})qFPR. \tag{12}$$

*where $p = \mathbb{P}(Y = 1)$ and $q = 1 - p$. Proof in Appendix C.2.*

From Corollary 3.2.1 we know that the index is decreasing in $\hat{p} = 1 - \lambda = \mathbb{P}(\hat{Y} = 1)$. According to Blackstone's[1] formulation, "It is better that ten guilty persons escape than that one innocent suffer." We can interpret this as meaning that the probability of a person being wrongfully convicted (FP)

---

[1]The famous British jurist, upon whose legal theories, the American legal system was built.

should be no more than one tenth of the probability that a guilty person escapes conviction (FN). That is, $\kappa FPR < FNR$. Let us denote the chosen ratio as $\kappa$, where $\kappa = 10$ corresponds to Blackstone's ratio. From Eq. (12) we know that

$$\frac{\mu - (q + pb_{TP})}{(1 - b_{TP})p} = FNR - \frac{(1 - b_{FP})q}{(1 - b_{TP})p}FPR > 0 \quad \Leftrightarrow \quad \mu > q + pb_{TP}$$

$$\text{where} \quad \kappa = \frac{(1 - b_{FP})q}{(1 - b_{TP})p} \quad \Rightarrow \quad b_- = b_{FP} = 1 - \frac{(1 - b_{TP})p\kappa}{q}. \tag{13}$$

Defining $\mu$ meaningfully, costs us one degree of freedom in the benefit matrix. We can satisfy Blackstones's constraint by simply ruling out all models for which $\mu < q + b_{TP}p$. If the index is strictly decreasing in $\mu$, capping the index where $\mu = q + pb_{TP}$, will have the desired effect. How can we ensure the index is decreasing in $\mu$ with the two remaining degrees of freedom ($b_{TP}$ and $\alpha$)? From Corollary 3.2.2, we know the index has a maxima where $\mu = \hat{\mu}(\lambda)$. If the turning point falls below the index domain, that is $\mu = \hat{\mu}(\lambda) < b_- + (1 - b_-)\lambda \Rightarrow I(\boldsymbol{b}; \alpha) \downarrow \mu$ and we have achieved the goal.

**3.2.2 Avoiding harm with assistive algorithms** In this example, the decision maker hires high scoring subjects. Thus, benefits should be increasing in $\hat{y}$. As regulator, we wish to avoid false negatives. Thus, $\lambda = \hat{p}$ and $(b_-, b_+) = (b_{FN}, b_{TN})$. From Eq. (4) we know that,

$$I(\boldsymbol{b}; \alpha) = \left[ (A_\alpha + \beta_\alpha)(1 - \hat{p}) + \beta_\alpha(\mu - 1) - f_\alpha(\mu) \right] / \mu^\alpha. \tag{14}$$

**Theorem 3.5** (Index as a function of the error rates for $\lambda = \hat{p}$). *For the benefit function $b_{ij} = ((b_{TN}, b_{FN}), (1, 1))$, where $b_{FN} < b_{TN} \in (0, 1)$, the index $I(\boldsymbol{b}; \alpha)$ can be written as a function of the false negative ($FNR$) and positive ($FPR$) rates,*

$$I(\boldsymbol{b}; \alpha) = \left[ pFNRf_\alpha(b_{FN}) + q(1 - FPR)f_\alpha(b_{TN}) - f_\alpha(\mu) \right] / \mu^\alpha \tag{15}$$

$$\mu = 1 - (1 - b_{TN})q(1 - FPR) - (1 - b_{FN})pFNR. \tag{16}$$

*where $p = \mathbb{P}(Y = 1)$, $q = 1 - p$. Proof in Appendix C.2.*

From Corollary 3.2.1 we know that the unfairness metric is increasing in the model reward rate $\hat{p}$. This time we wish to avoid false negatives $\kappa FNR < FPR$. As before defining the mean benefit meaningfully costs us a degree of freedom. From Eq. (16) we know that

$$\frac{\mu - (p + qb_{TN})}{(1 - b_{TN})} = FPR - \frac{(1 - b_{FN})p}{(1 - b_{TN})q}FNR > 0 \quad \Leftrightarrow \quad \mu > p + qb_{TN}.$$

$$\text{where} \quad \kappa = \frac{(1 - b_{FN})p}{(1 - b_{TN})q} \quad \Rightarrow \quad b_- = b_{FN} = 1 - \frac{(1 - b_{TN})q\kappa}{p}, \tag{17}$$

We want to rule out models for which $\mu < p + b_{TN}q$. If the index is strictly decreasing in $\mu$, capping the index where $\mu = p + qb_{TN}$ has the desired effect. If the turning point falls below the index domain $I(\boldsymbol{b}; \alpha) \downarrow \mu$, this must be the case.

**Summary** From Eqs. (13) and (17) we know that $b_+$ and $b_-$ are related, leaving only one degree of freedom in the benefit matrix. For brevity, we denote,

$$b_- = 1 - (1 - b_+)\tilde{\kappa} \quad \Rightarrow \quad 0 < b_- < 1 - 1/\tilde{\kappa} < b_+ \tag{18}$$

where $\tilde{\kappa} = \kappa p/q$ or $\tilde{\kappa} = \kappa q/p$ depending on whether we wish to avoid false positives or negatives respectively. Substituting for $b_-$ in the equations allows us to drop the benefit subscript. We can write the both the difference $\delta = b_+ - b_-$ and ratio $\varphi = b_-/b_+$ in terms of $b_+ = b$.

$$\delta = (\tilde{\kappa} - 1)(1 - b) \quad \text{and} \quad \varphi = 1 - \delta/b = 1 - \epsilon \quad \text{where} \quad \epsilon = \delta/b = (\tilde{\kappa} - 1)(1 - b)/b. \tag{19}$$

Note that $\delta, \varphi \in (0, 1)$.

**3.2.3 Ensuring the index is monotonic in the mean benefit** Intuitively, we know that if $b_-$ is close to zero and $b_+$ is close to one, this will teach the algorithm to avoid whichever error type is assigned the benefit $b_-$. In general, a regulator must prioritize the errors a greedy decision maker will ignore. We have one degree of freedom left, in the benefit matrix. A natural limit to set as a

regulator is the minimum benefit $b_-$. In law we already employ the concept of a minimum legal benefit which guarantees a reasonable minimum information exchange from decision makers to subject. In many countries and some US states such as California, there is a requirement that salary bands are shared on all job postings. An an entirely reasonable piece of information that potential candidates should have, to enable them to filter job postings. Similarly, some jurisdictions require a *reason* to be provided to the applicant, when a loan is rejected. The minimum benefit increases with transparency - it saves the masses time and provides them with the opportunity to rectify erroneous information about them. These provide examples of policies which decision makers can implement to raise the minimum benefit $b_-$ in their benefit matrix. The question is only one around how to communicate the value of a policy for a given application.

We still need to choose $\alpha$. Ideally we would make some sensible choice of $\alpha$, and then use the remaining benefit to ensure the index is monotonic. What is a sensible choice of $\alpha$? We know that when minimizing risk, $\alpha = 0$ provides a well reasoned choice and is a natural starting point for investigation. That said, given only binary arrays, we must choose $\alpha > 0$. We also know from Section 2, that values of $\alpha \in (0, 1)$, discounts the total contribution from the within group component, such that the discount is greatest when $\alpha = 1/2$. Looking at Eq. (2), we see that for a group $g$, the contribution of the group $I(\boldsymbol{b}_g)$ to the within-group component $I_\omega^G(\boldsymbol{b})$ of the index, is multiplied by a factor of $(\mu_g/\mu)^\alpha$, let's call this the *grit factor*. Like a *discount factor*, which is applied to a future cashflow to calculate its *present value*, the grit factor adjusts the within-group contribution from group $g$, $I(\boldsymbol{b}_g)$. Unlike the discount factor, the grit factor discounts some scores and inflates others. We can see that the grit factor is always one for mean scoring groups.

We can calculate

$$e^{-r} = (\mu_g/\mu)^\alpha \quad \Rightarrow \quad r = \alpha \ln(\mu/\mu_g) = \alpha[\ln \mu - \ln \mu_g]$$

is the *continually compounding* interest rate, or *grit rate*, on a future cashflow of $I(\boldsymbol{b}_g)$ and is proportional to $\alpha$. The grit rate is always zero for mean scoring groups, it is positive for $\mu_g < \mu$, and negative for $\mu_g > \mu$. Like the mean (Eq. (6)), the group mean $\mu_g \in [b_-, 1]$ and consequently the grit rate $r \in [\alpha \ln(\mu), \alpha \ln(\mu/b_-)]$ are bounded.

If that each individual has access to the average utility of their peers, values of $\alpha$ a adjustment factor to reflect its *actual value*, assuming the test is biased. The grit factor inflates low scores, and discounts high scores proportionally. Eq. (2) shows that the size of a subgroup, greatly influences its contribution to the between-group component $I_\beta^G$ of the inequality index. We know that variance scales linearly. When we calculate the mean benefit of each group, we divide by the number of observations in the group $n_g$, which reduces the variance by $1/n_g$, and mean estimation error by $1/\sqrt{n_g}$. Taking the square root, after calculating the mean replaces the lost variance in $\mu_g/\mu \, I_\beta^G$, thus ensuring all groups (regardless of size) contribute the same variance, ultimately accounting for representation bias.

## 4  DISCUSSION

The findings of this paper and the works which led to it affect all of us, several times a day. Every time we make a judgment about a person (especially an emotive one), how can we be less biased? From prior works we know that using a conventional (both evidence based and biased) model of utility with a binary outcome, a decision maker cannot be fair Friedler et al. (2016). From Barocas et al. (2019), we know that introducing a third possible outcome (increasing the size of our outcome space from binary to ternary), makes satisfying *independence* ($\hat{Y} \perp Z$) and *separation* ($\hat{Y} \perp Z|Y$) possible. What third outcome could there possibly be? Surely, someone either is or it isn't something? No. There is *always* another possibility. More fundamentally, this result says that, *any* scale on which we measure or represent people, cannot be binary.

There is one remaining degree of freedom in the generalization parameter $\alpha$. In all societies, *personal* and *social wealth* tend to be correlated. Here we define personal and social wealth to be $b_i/\mu$ and $\mu_g/\mu$ respectively in Eq. (2). A targeted advertising algorithm for luxury goods, would indeed need to be a good predictor of how wealthy an individual is, and *rich people tend to have rich friends*, so a true value of $\alpha > 0$ makes sense, but *rich $\neq$ will purchase* thus $\alpha < 1$. In truth, the target $Y$ is generally a positively correlated proxy for the thing we (as the decision maker) would like to measure $\tilde{Y}$. By choosing $\alpha = 0$, we ensure the model $\hat{Y}$ is an unbiased estimator of $Y$, but

$Y$ isn't really what we want, $\tilde{Y}$ is. If our test $Y$ is biased toward some dimension $Z$, then we would expect the differences $Y - \tilde{Y} = \epsilon(Z)$ to be increasing in $Z$. The greater the reliance on the proxy $Y$, the more exaggerated the bias. In theory if we could estimate $\alpha$, and account for it; resulting in a more accurate measure of utility.

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

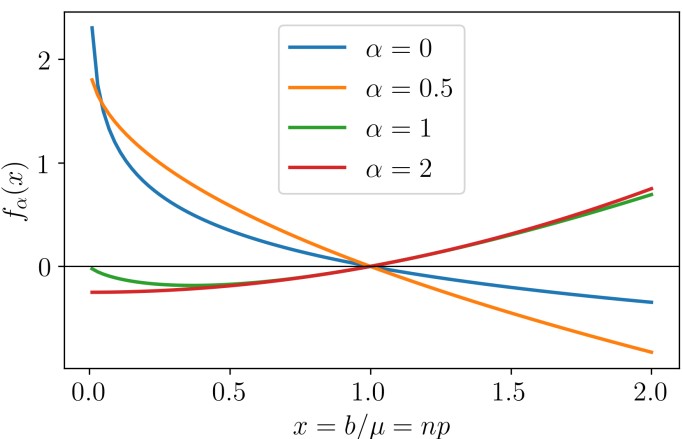

Figure 2: $f_\alpha(x)$ for varying $\alpha$.

# A   GENERALIZED ENTROPY INDICES

$$I(\boldsymbol{b}; \alpha) = \frac{1}{n} \sum_{i=1}^{n} f_\alpha \left( \frac{b_i}{\mu} \right), \quad f_\alpha(x) = \begin{cases} -\ln x & \text{for } \alpha = 0 \\ x \ln x & \text{for } \alpha = 1 \\ \dfrac{x^\alpha - 1}{\alpha(\alpha - 1)} & \text{o.w.} \end{cases} \tag{1}$$

Differentiating $f_\alpha(x)$ in equation (1) with respect to $x$ gives,

$$f_\alpha'(x) = \begin{cases} 1 + \ln x & \text{for } \alpha = 1 \\ x^{\alpha-1}/(\alpha - 1) & \text{otherwise,} \end{cases} \tag{20}$$

and $f_\alpha''(x) = x^{\alpha-2} > 0 \ \forall \ \alpha, \ x$. Thus, $f_\alpha(x)$ is convex.

In Fig. 2, we plot the function $f_\alpha(x)$, for different choices of $\alpha$.

## A.1   INTEGRAND BEHAVIOR

**Theorem A.1** (Behavior of the Integrand).

$$\alpha < 1 \ \Rightarrow \ f_\alpha(x) \text{ is strictly decreasing.}$$
$$\alpha = 1 \ \Rightarrow \ f_\alpha(x) \text{ is minimal at } x = e^{-1}.$$
$$\alpha > 1 \ \Rightarrow \ f_\alpha(x) \text{ is strictly increasing.}$$

*Proof.* For $\alpha = 0$,

$$f_0(x) = -\ln(x) \quad \Rightarrow \quad f_0'(x) = -\frac{1}{x} < 0 \quad \text{for} \quad x > 0$$
$$\Rightarrow \quad f_0(x) \text{ strictly decreasing for } x > 0$$
$$f_0(x) = 0 \quad \Leftrightarrow \quad x = 1.$$

For $\alpha = 1$,

$$f_1(x) = x \ln x \quad \Rightarrow \quad f_1'(x) = 1 + \ln x = 0 \quad \Leftrightarrow \quad x = \frac{1}{e}.$$

$$\Rightarrow \quad f_1''(x) = \frac{1}{x} > 0 \quad \forall\, x > 0$$

$$\Rightarrow \quad f_1(x) \text{ is minimal at } x = \frac{1}{e}$$

$$f_1(x) = 0 \quad \Leftrightarrow \quad x \in \{0, 1\},$$

$$\Rightarrow \quad f_1(x) > 0 \text{ for } x > 1 \quad \text{and}$$

$$f_1(x) < 0 \text{ for } x < 1$$

For $\alpha \notin \{0, 1\}$,

$$f_\alpha(x) = \frac{x^\alpha - 1}{\alpha(\alpha - 1)} \quad \Rightarrow \quad f_\alpha'(x) = \frac{x^{\alpha-1}}{\alpha - 1}.$$

$$\Rightarrow \quad f_1'(x) > 0 \text{ if } \alpha > 1 \quad \text{and}$$

$$f_1'(x) < 0 \text{ if } \alpha < 1$$

$$\Rightarrow \quad f_\alpha(x) \text{ strictly decreasing for } \alpha < 1$$

$$\Rightarrow \quad f_\alpha(x) \text{ strictly increasing for } \alpha > 1$$

$\square$

## A.2   RELATIONSHIP WITH RISK

**Theorem 3.1** (Influence of the Mean Benefit on the Generalized Entropy Index)

$$I(\boldsymbol{b}; \alpha) = \mathbb{E}\left[ f_\alpha \left( \frac{b}{\mu} \right) \right] = \frac{\mathbb{E}[f_\alpha(b)] - f_\alpha(\mu)}{\mu^\alpha}, \tag{3}$$

where $f_\alpha$ is defined in equation (1).

*Proof.* From equation (1),

$$\alpha = 0 \quad \Rightarrow \quad I(\boldsymbol{b}; \alpha) = \mathbb{E}\left[ -\ln\left( \frac{B}{\mu} \right) \right] = \mathbb{E}[-\ln B] + \ln \mu.$$

$$\alpha = 1 \quad \Rightarrow \quad I(\boldsymbol{b}; \alpha) = \mathbb{E}\left[ \frac{B}{\mu} \ln\left( \frac{B}{\mu} \right) \right] = \frac{1}{\mu}\mathbb{E}[B \ln B - B \ln \mu]$$

$$= \frac{1}{\mu}[\mathbb{E}(B \ln B) - \mu \ln \mu].$$

$$\alpha \notin \{0, 1\} \quad \Rightarrow \quad I(\boldsymbol{b}; \alpha) = \frac{1}{\alpha(\alpha - 1)}\mathbb{E}\left[ \left( \frac{B}{\mu} \right)^\alpha - 1 \right] = \frac{\mathbb{E}(B^\alpha) - \mu^\alpha}{\alpha(\alpha - 1)\mu^\alpha}$$

$$= \frac{\mathbb{E}(B^\alpha) - 1 - (\mu^\alpha - 1)}{\alpha(\alpha - 1)\mu^\alpha}.$$

$\square$

## B   INDEX FOR TWO BENEFIT LEVELS

Here we consider the simplest case where there is one degree of freedom (or equivalently two benefit levels) in the matrix, a high benefit and a low benefit. Since the index is scale invariant we can choose one of these to be unity, and denote the other benefit as $b \geq 0$. For known $b$, the benefit distribution (and thus index behavior) can be characterised with a single variable, the proportion of individuals receiving unit benefit, which we denote $\rho$. In theory, any of the elements $b_{ij}$ could be unity, leading

to a different interpretation of $\rho$. Two important cases include, model accuracy (if the diagonal $\hat{y} = y$ results in unit benefit) and the acceptance rate (if the row $\hat{y} = 1$ results in unit benefit).

Note that for a benefit matrix with more than two benefit levels, all benefit matrices fall into one of two types. Either one of the diagonals dominates, or one of the rows dominates. If a diagonal dominates we can reasonably assume it is the leading diagonal (accurate predictions being more beneficial than errors). If a row dominates, we can assume without loss of generality that it is positive predictions that are most beneficial. We consider the simplest case, $b = 0$ first.

## B.1 BINARY BENEFITS

**Theorem B.1** (Index Behavior for Binary Benefits). *For a binary benefit array $\boldsymbol{b}$, with mean benefit $\mu \in (0, 1]$, the index $I(\boldsymbol{b}; \alpha)$ is a strictly decreasing function of the mean benefit. See appendix B.1 for the proof.*

*Proof.* For binary benefits, with mean benefits $\mu$, $n\mu$ of the $n$ individuals receive a benefit of one and the remaining $n(1 - \mu)$ receive a benefit of zero. Thus, we can write the value of the index as,

$$I(\boldsymbol{b}; \alpha) = (1 - \mu)f_\alpha(0) + \mu f_\alpha\left(\frac{1}{\mu}\right),$$

Substituting in the index yields,

$$I(\boldsymbol{b}; \alpha) = \begin{cases} -\ln \mu & \text{for } \alpha = 1 \\ \dfrac{\mu^{1-\alpha} - 1}{\alpha(\alpha - 1)} & \text{for } \alpha > 0. \end{cases} \tag{21}$$

Note that for binary benefits we must have $\alpha > 0$. From equation (21), for $\alpha = 1$ it is straightforward to see that the index is decreasing in $\mu$. For $\alpha > 1$, the exponent of $\mu$ is negative. For $\alpha \in (0, 1)$, the exponent is positive but the denominator is negative. $\square$

Theorem B.1 tells us that for binary benefits, the generalized entropy index is a monotonic decreasing function of the mean benefit, regardless of the choice of $\alpha$. The value of applying inequality indices for binary benefits then is questionable, since the index calculation introduces a free parameter $\alpha$, and the index value is far more opaque in meaning than the mean benefit itself. For $b_{ij} = ((1, 0), (0, 1))$, the mean benefit $\mu$ is exactly the model accuracy and the index ranks the fairness of models in order of accuracy. For $b_{ij} = ((0, 0), (1, 1))$, the mean benefit $\mu$ is exactly the acceptance rate and the index ranks the fairness of models in order of acceptance rate. For binary benefits, the only one way to achieve equality is if all individuals receive a benefit of one, since the index is undefined for $\mu = 0$; when $b > 0$, this is no longer the case and there are two ways to achieve equality in benefits.

## B.2 TWO NON-ZERO BENEFIT LEVELS

**Theorem B.2** (Index Behavior for Two Benefit Levels). *Let $\rho$ be the proportion of individuals which receive unit benefit, and $b$ be the benefit the remaining individuals receive; the index $I(\boldsymbol{b}; \alpha)$ is zero for $\rho = 1$. For $b > 0$, the index is also zero for $\rho = 0$, and takes its maximal value for some $\rho = \hat{\rho}(b, \alpha) \in (0, 1)$.*

$$\alpha < -1 \;\Rightarrow\; \hat{\rho}(b, \alpha) \downarrow b,$$
$$\alpha = -1 \;\Rightarrow\; \hat{\rho}(b, \alpha) = 1/2$$
$$\text{and} \qquad \alpha > -1 \;\Rightarrow\; \hat{\rho}(b, \alpha) \uparrow b.$$

$$\hat{\rho}(b, \alpha) \to \left\{ \begin{array}{cc} 0 & \text{for } \alpha \geq 0 \\ \alpha/(\alpha - 1) & \text{for } \alpha < 0 \end{array} \right\} \text{ as } b \to 0$$

$$\text{and} \qquad \hat{\rho}(b, \alpha) \to \left\{ \begin{array}{cc} 1 & \text{for } \alpha \geq 0 \\ 1/(1 - \alpha) & \text{for } \alpha < 0 \end{array} \right\} \text{ as } b \to \infty.$$

$$0 < b < 1 \;\Rightarrow\; \hat{\rho}(b, \alpha) \uparrow \text{ in } \alpha$$
$$\text{and} \qquad b > 1 \;\Rightarrow\; \hat{\rho}(b, \alpha) \downarrow \text{ in } \alpha.$$

*Proof.* From equation (3), since $f_\alpha(1) = 0$, we can write the value of the index as,

$$I(\boldsymbol{b}; \alpha) = \frac{(1 - \rho)f_\alpha(b) - f_\alpha(\mu)}{\mu^\alpha}.$$

For symmetric benefits, with mean benefits $\mu$, of $n$ individuals, $n\rho$ receive a benefit of one and the remaining $n(1 - \rho)$ receive a benefit $b$. Thus,

$$\mu = \rho + (1 - \rho)b \quad \Leftrightarrow \quad \mu = (1 - b)\rho + b$$
$$\Leftrightarrow \quad \rho = \frac{\mu - b}{1 - b} \quad \Leftrightarrow \quad 1 - \rho = \frac{1 - \mu}{1 - b}.$$

Substituting for $1 - \rho$ in the expression for the index above gives,

$$I(\boldsymbol{b}; \alpha) = \frac{(1 - \mu)f_\alpha(b) - (1 - b)f_\alpha(\mu)}{(1 - b)\mu^\alpha}.$$

Substituting for $f_\alpha$ from equation (1) yields,

$$I(\boldsymbol{b}; \alpha) = \begin{cases} \ln\mu - \dfrac{(1 - \mu)\ln b}{1 - b} & \text{for } \alpha = 0 \\[2ex] \dfrac{b\ln b}{1 - b}\left(\dfrac{1}{\mu} - 1\right) - \ln\mu & \text{for } \alpha = 1 \\[2ex] \dfrac{(1 - \mu)(b^\alpha - 1) - (1 - b)(\mu^\alpha - 1)}{\alpha(\alpha - 1)(1 - b)\mu^\alpha} & \text{o.w.} \end{cases}$$

Rearranging gives,

$$I(\boldsymbol{b}; \alpha) = \begin{cases} \ln\mu - \dfrac{(1 - \mu)\ln b}{1 - b} & \text{for } \alpha = 0 \\[2ex] \dfrac{b\ln b}{1 - b}\left(\dfrac{1}{\mu} - 1\right) - \ln\mu & \text{for } \alpha = 1 \\[2ex] \dfrac{b(b^{\alpha-1} - 1) - (b^\alpha - 1)\mu - (1 - b)\mu^\alpha}{\alpha(\alpha - 1)(1 - b)\mu^\alpha} & \text{o.w.} \end{cases} \tag{22}$$

Differentiating equation (22) with respect to $\mu$,

$$\frac{\mathrm{d}I}{\mathrm{d}\mu} = \begin{cases} \dfrac{1}{\mu} + \dfrac{\ln b}{1-b} \\[2ex] \dfrac{b\ln b}{(1-b)\mu^2} + \dfrac{1}{\mu} \\[2ex] \dfrac{\alpha b(1-b^{\alpha-1}) - (\alpha-1)(1-b^\alpha)\mu}{\alpha(\alpha-1)(1-b)\mu^{\alpha+1}} \end{cases}$$

$$\frac{\mathrm{d}I}{\mathrm{d}\mu} = 0 \quad \Leftrightarrow \quad \mu = \hat{\mu} = \begin{cases} \dfrac{b-1}{\ln b} & \text{for } \alpha = 0 \\[2ex] \dfrac{b\ln b}{b-1} & \text{for } \alpha = 1 \\[2ex] \dfrac{\alpha b(b^{\alpha-1}-1)}{(\alpha-1)(b^\alpha-1)} & \text{o.w.} \end{cases}$$

$$\Leftrightarrow \quad \hat{\rho}(b,\alpha) = \frac{b-\hat{\mu}}{b-1} = \begin{cases} \dfrac{b}{b-1} - \dfrac{1}{\ln b} & \text{for } \alpha = 0 \\[2ex] \dfrac{b}{b-1}\left(1 - \dfrac{\ln b}{b-1}\right) & \text{for } \alpha = 1 \\[2ex] \dfrac{b}{b-1}\left(1 - \dfrac{\alpha(b^{\alpha-1}-1)}{(\alpha-1)(b^\alpha-1)}\right) & \text{o.w.} \end{cases}$$

$$\Leftrightarrow \quad \hat{\rho}(b,\alpha) = \begin{cases} \dfrac{b\ln b - (b-1)}{(b-1)\ln b} & \text{for } \alpha = 0 \\[2ex] \dfrac{b(b-1-\ln b)}{(b-1)^2} & \text{for } \alpha = 1 \\[2ex] \dfrac{b[(\alpha-1)(b^\alpha-1) - \alpha(b^{\alpha-1}-1)]}{(\alpha-1)(b-1)(b^\alpha-1)} & \text{o.w.} \end{cases}$$

$$\Leftrightarrow \quad \hat{\rho}(b,\alpha) = \begin{cases} \dfrac{b\ln b - (b-1)}{(b-1)\ln b} & \text{for } \alpha = 0 \\[2ex] \dfrac{b(b-1-\ln b)}{(b-1)^2} & \text{for } \alpha = 1 \\[2ex] \dfrac{b[(\alpha-1)b^\alpha - \alpha b^{\alpha-1} + 1]}{(\alpha-1)(b-1)(b^\alpha-1)} & \text{o.w.} \end{cases} \qquad (23)$$

We plot $\hat{\rho}(b,\alpha)$ as a function of $b$ for varying $\alpha$ in Figure 3.

BEHAVIOR WITH RESPECT TO $\alpha$

BEHAVIOR WITH RESPECT TO $b$

We want to know the location of the index maxima $\hat{\rho}(b,\alpha)$ in the extremes when $b = 0$ and $b = 1$. Finding the behavior of $\hat{\rho}(b,\alpha)$ as $b \to 0$ is straightforward.

$$\hat{\rho}(b,\alpha) \to \left\{ \begin{array}{ll} 0 & \text{for } \alpha \geq 0 \\ \alpha/(\alpha-1) & \text{for } \alpha < 0 \end{array} \right\} \quad \text{as} \quad b \to 0 \qquad (24)$$

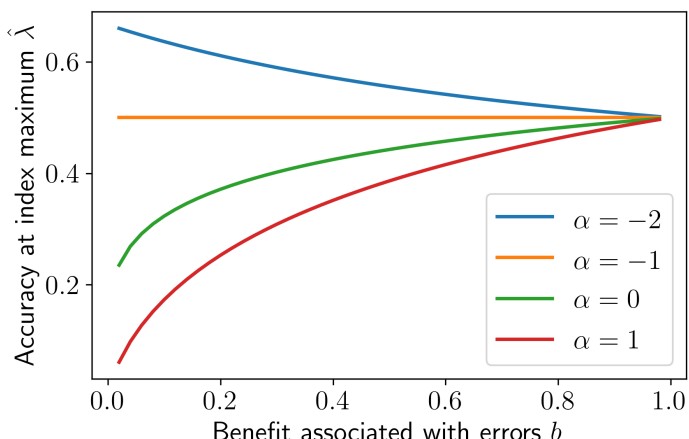

Figure 3: $\hat{\rho}(b, \alpha)$ as a function of $b$ for varying $\alpha$. See equation (22)

From equation (23), we can see that for $b = 1$, $\hat{\rho}(b, \alpha)$ is indeterminate. Applying l'Hopital's rule gives,

$$\lim_{b \to 1} \{\hat{\rho}(b, \alpha)\} = \begin{cases} \dfrac{\ln b}{1 + \ln b - 1/b} & \text{for } \alpha = 0 \\[2mm] \dfrac{2(b-1) - \ln b}{2(b-1)} & \text{for } \alpha = 1 \\[2mm] \dfrac{(\alpha-1)(\alpha+1)b^{\alpha} - \alpha^2 b^{\alpha-1} + 1}{(\alpha-1)[(\alpha+1)b^{\alpha} - \alpha b^{\alpha-1} - 1]} & \text{o.w.,} \end{cases}$$

$$= \begin{cases} \dfrac{1/b}{1/b + 1/b^2} & \text{for } \alpha = 0 \\[2mm] \dfrac{2 - 1/b}{2} & \text{for } \alpha = 1 \\[2mm] \dfrac{\alpha(\alpha+1)b^{\alpha-1} - \alpha^2 b^{\alpha-2}}{\alpha(\alpha+1)b^{\alpha-1} - \alpha(\alpha-1)b^{\alpha-2}} & \text{o.w.,} \end{cases}$$

shows that

$$\hat{\rho}(b, \alpha) = 1/2 \quad \text{for } b = 1 \quad \forall \, \alpha. \tag{25}$$

We note that if $b > 1$ then the unit benefit no longer dominates. Since the index is scale invariant, dividing all benefits by $b$, does not change the value of the index; but then $1 - \rho$ (as oppose to $\rho$) of the individuals receive unit benefit and the remaining $\rho$ receive a benefit of $1/b$. Thus,

$$\lim_{b \to \infty} \{\hat{\rho}(b, \alpha)\} = \lim_{b \to \infty} \{1 - \hat{\rho}(1/b, \alpha)\} = 1 - \lim_{b \to 0}\{\hat{\rho}(b, \alpha)\}.$$

From equation (24)

$$\hat{\rho}(b, \alpha) \to \begin{cases} 1 & \text{for } \alpha \geq 0 \\ 1/(1 - \alpha) & \text{for } \alpha < 0 \end{cases} \quad \text{as} \quad b \to \infty \tag{26}$$

$\square$

Theorem B.2 tells us that, when $b > 0$ both $\rho = 0$ and $\rho = 1$ are *perfectly* fair models for which the index value is zero. In the case where accurate predictions dominate the benefit matrix, this corresponds to a model accuracy of zero or one. That said, for any reasonable binary classifier, the model accuracy must be greater than 1/2, and since for $\alpha \geq -1$, the maxima occurs for a model accuracy of less than 1/2, the index $I(b; \alpha)$ remains a decreasing function of model accuracy for any reasonable model. In the case where positive predictions dominate, these two perfectly fair scenarios correspond to rejecting everyone or accepting everyone.

## C INDEX FOR THREE BENEFIT LEVELS

### C.1 TWO NON-ZERO BENEFIT LEVELS

**Theorem C.1** (Index for $b_- = 0$). *Given the benefit matrix*

$$b_{ij} = \begin{pmatrix} 1 & 0 \\ b_+ & 1 \end{pmatrix}$$

*the generalized entropy index is defined for $\alpha > 0$ and can be written as:*

$$I(\mu, \lambda) = \begin{cases} \left(1 - \dfrac{\lambda}{\mu}\right) \ln b_+ - \ln \mu & \text{for } \alpha = 1 \\[4mm] \dfrac{1}{\alpha(\alpha-1)} \left[ \left(\dfrac{b_+}{\mu}\right)^{\alpha-1} - \dfrac{(b_+^{\alpha-1} - 1)}{\mu^\alpha} \lambda - 1 \right] & \text{for } \alpha > 0 \end{cases}$$

*Proof.* Let's suppose the model makes $n_c$ correct predictions (in which case $b = 1$); $n_+$ false positive predictions (in which case $b = b_+$); and the remaining $n - n_c - n_+$ predictions are false negative (in which case $b = 0$). We can write the value of the index as,

$$I(\boldsymbol{b}; \alpha) = \frac{1}{n} \left[ (n - n_c - n_+) f_\alpha(0) + n_c f_\alpha\left(\frac{1}{\mu}\right) + n_+ f_\alpha\left(\frac{b_+}{\mu}\right) \right].$$

Using equation (1) we can show that,

$$I(\boldsymbol{b}; \alpha) = \begin{cases} -\dfrac{(n_c + b_+ n_+)}{n} \dfrac{\ln \mu}{\mu} + \dfrac{b_+ n_+ \ln b_+}{n\mu} & \text{for } \alpha = 1 \\[4mm] \dfrac{1}{\alpha(\alpha-1)} \left( \dfrac{n_c + b_+^\alpha n_+}{n\mu^\alpha} - 1 \right) & \text{for } \alpha > 0. \end{cases}$$

Let us denote the model accuracy with $\lambda$. We have,

$$\lambda = \frac{n_c}{n} \quad \text{and} \quad \mu = \frac{n_c + b_+ n_+}{n} \quad \Rightarrow \quad \frac{b_+ n_+}{n} = \mu - \lambda.$$

Substituting completes the proof. $\qquad\square$

**Theorem C.2** (Index turning point). *The index has exactly one turning point for $\alpha > 0$ at $\mu = \hat{\mu}$ where, $\hat{\mu} = g(b_+, \alpha)\lambda$ and,*

$$g(b_+, \alpha) = \begin{cases} \ln b_+ & \text{for } \alpha = 1 \\[3mm] \dfrac{\alpha(b_+^{\alpha-1} - 1)}{(\alpha-1)b_+^{\alpha-1}} & \text{for } \alpha > 0 \end{cases}$$

*The stationary point is an inflection point if $b_+ = 1$, a minima if $b_+ < 1$, and maxima if $b_+ > 1$.*

*Proof.* Differentiating equation (4),

$$\frac{\partial I}{\partial \mu} = \begin{cases} \dfrac{1}{\mu^2} (\lambda \ln b_+ - \mu) & \text{for } \alpha = 1 \\[4mm] \dfrac{\alpha(b_+^{\alpha-1} - 1)\lambda - (\alpha-1)b_+^{\alpha-1}\mu}{\alpha(\alpha-1)\mu^{\alpha+1}} & \text{for } \alpha > 0 \end{cases}$$

$$\Rightarrow \quad \frac{\partial I}{\partial \mu} = 0 \quad \Leftrightarrow \quad \mu = \hat{\mu} = g(\alpha)\lambda.$$

Differentiating again,

$$
\frac{\partial^2 I}{\partial \mu^2} =
\begin{cases}
\dfrac{1}{\mu^3}\left[\mu - 2\lambda \ln b_+\right] & \text{for } \alpha = 1 \\[2ex]
\dfrac{b_+^{\alpha-1}}{\mu^{\alpha+2}}\left[\mu - \dfrac{(\alpha+1)(b_+^{\alpha-1}-1)}{(\alpha-1)b_+^{\alpha-1}}\lambda\right] & \text{for } \alpha > 0
\end{cases}
$$

$$
\Rightarrow \quad \frac{\partial^2 I}{\partial \mu^2}\bigg|_{\mu=\hat{\mu}} =
\begin{cases}
-\dfrac{\ln b_+}{\hat{\mu}^3}\lambda & \text{for } \alpha = 1 \\[2ex]
-\dfrac{(b_+^{\alpha-1}-1)}{\hat{\mu}^{\alpha+2}(\alpha-1)}\lambda & \text{for } \alpha > 0
\end{cases}
$$

$$
\Rightarrow \quad \frac{\partial^2 I}{\partial \mu^2}\bigg|_{\mu=\hat{\mu}}
\begin{cases}
> 0 & \text{for } b_+ < 1 \\
= 0 & \text{for } b_+ = 1 \\
< 0 & \text{for } b_+ > 1
\end{cases}
\quad \forall\, \alpha > 0.
$$

$\square$

**Theorem C.3** (The Deviation Region).

$$
\left.
\begin{array}{lll}
\Delta I^-(\mu,\lambda;n) < 0 & \Rightarrow & \mu < h^-(b_+,\alpha)\lambda \\
\Delta I^+(\mu,\lambda;n) < 0 & \Rightarrow & \mu > h^+(b_+,\alpha)\lambda
\end{array}
\right\} \tag{27}
$$

*where,*

$$
h^\pm(b_+,\alpha) =
\begin{cases}
\dfrac{(b_+-1)\ln b_+}{b_+ - 1 \mp \ln b_+} & \text{if} \quad \alpha = 1 \\[3ex]
\dfrac{\alpha(b_+-1)(b_+^{\alpha-1}-1)}{[(\alpha-1)(b_+-1)\mp 1]b_+^{\alpha-1}\pm 1} & \text{if} \quad \alpha > 0,\ \alpha \neq 1.
\end{cases} \tag{28}
$$

*Proof.* Eq. (4) provides an expression for $I(\mu,\lambda)$. Substituting for $\lambda$ and $\mu$ in the case $\alpha = 1$ gives,

$$
I\left(\mu \pm \frac{\delta}{n}, \lambda - \frac{1}{n}\right) = \left[1 - \left(\frac{\lambda}{\mu} - \frac{1}{n\mu}\right)\left(1 \pm \frac{\delta}{n\mu}\right)^{-1}\right]\ln b_+
$$

$$
- \ln \mu - \ln\left(1 \pm \frac{\delta}{n\mu}\right).
$$

For $\alpha > 0$, we get,

$$
I\left(\mu \pm \frac{\delta}{n}, \lambda - \frac{1}{n}\right) = \frac{1}{\alpha(\alpha-1)}\left[\left(\frac{b_+}{\mu}\right)^{\alpha-1}\left(1 \pm \frac{\delta}{n\mu}\right)^{1-\alpha}\right.
$$

$$
\left. - \frac{(b_+^{\alpha-1}-1)}{\mu^{\alpha-1}}\left(\frac{\lambda}{\mu} - \frac{1}{n\mu}\right)\left(1 \pm \frac{\delta}{n\mu}\right)^{-\alpha} - 1\right].
$$

We showed earlier that we must have, $\lambda \leq \mu \leq b_+ + (1 - b_+)\lambda$, in addition, any reasonable model should satisfy $0.5 \leq \lambda \leq 1$. We deduce that we must have $0.5 \leq \mu \leq b_+ + 0.5$ and so $\mu = \mathrm{O}(1)$. Then for large $n$, we can be sure that $n\mu$ is large and its reciprocal $\epsilon = 1/(n\mu)$ is small. For large $n$, we can write the cost of an error as

$$
\Delta I_\alpha^\pm(\mu,\lambda;n) = \xi_\alpha(\mu,\lambda)\epsilon + \mathrm{O}(\epsilon^2)
$$

where,

$$
\xi_\alpha(\mu,\lambda) =
\begin{cases}
\left(1 \pm \dfrac{\delta\lambda}{\mu}\right)\ln b_+ \mp \delta & \text{for } \alpha = 1 \\[3ex]
\dfrac{[[1 \pm (1-\alpha)\delta]b_+^{\alpha-1}-1]\mu \pm \alpha\delta(b_+^{\alpha-1}-1)\lambda}{\alpha(\alpha-1)\mu^{\alpha-1}} & \text{for } \alpha > 0.
\end{cases}
$$

$\square$

## C.2 THREE NON-ZERO BENEFIT LEVELS

**Theorem 3.3** (Index as a function of the error distribution when $\lambda = \mathbb{P}(\hat{Y} = Y)$). *For the benefit function $b_{ij} = ((1, b_-), (b_+, 1))$, the index $I(\boldsymbol{b}; \alpha)$ can be written as a function of the false positive and false negative rates, $FPR$ and $FNR$ respectively,*

$$I(\boldsymbol{b}; \alpha) = [f_\alpha(b_-)pFNR + f_\alpha(b_+)qFPR - f_\alpha(\mu)]/\mu^\alpha \tag{8}$$

$$\mu = 1 + (b_- - 1)pFNR + (b_+ - 1)qFPR. \tag{9}$$

*where $p = \mathbb{P}(Y = 1)$ and $q = 1 - p$.*

*Proof.* Let the proportion of accurate, false negative and false positive predictions be denoted by $\lambda$, $p_-$ and $p_+$ respectively. Since $f_\alpha(1) = 0 \; \forall \; \alpha$, from equation (3) we know

$$I(\boldsymbol{b}; \alpha) = [p_- f_\alpha(b_-) + p_+ f_\alpha(b_+) - f_\alpha(\mu)]/\mu^\alpha. \tag{29}$$

where $p_\pm$ is the probability of the benefit $b_{\pm}$. Note that, any of the elements $b_{ij}$ could be assigned one of the three benefits, and not affect the validity of this representation. We also know,

$$\lambda + p_- + p_+ = 1, \tag{30}$$

and given the mean benefit $\mu$,

$$\mu = \lambda + b_- p_- + b_+ p_+. \tag{31}$$

We can use equation (30) to eliminate $\lambda$ from equation (31) giving,

$$\mu = 1 + (b_- - 1)p_- + (b_+ - 1)p_+. \tag{32}$$

For convenience we write our probability matrix in terms of the subject relevant errors:

$$\mathbb{P}(\hat{y} = i, y = j) = \begin{pmatrix} q(1 - FPR) & pFNR \\ qFPR & p(1 - FNR) \end{pmatrix}. \tag{33}$$

$$\lambda = \mathbb{P}(\hat{Y} = Y), \quad p_+ = qFPR \quad \text{and} \quad p_- = pFNR.$$

Substituting into equations (29) and (32) completes the proof. $\qquad \square$

**Theorem 3.2** (Index as a function of $\lambda$ and $\mu$) *For the benefits $b_i \in \{b_-, b_+, 1\}$, where $b_- < b_+$ the index $I(\boldsymbol{b}; \alpha)$ can be written as a function of the mean benefit $\mu$ and the unit reward rate $\lambda$*

$$I(\boldsymbol{b}; \alpha) = [(A_\alpha + \beta_\alpha)(1 - \lambda) + \beta_\alpha(\mu - 1) - f_\alpha(\mu)]/\mu^\alpha \tag{4}$$

*where $A_\alpha = f_\alpha(b_+) - b_+ \beta_\alpha$ and $\beta_\alpha = [f_\alpha(b_+) - f_\alpha(b_-)]/(b_+ - b_-)$. $A_\alpha$ and $\beta_\alpha$ are respectively the intercept and the gradient of the straight line passing through $(b_-, f_\alpha(b_-))$ and $(b_+, f_\alpha(b_+))$.*

*Proof.* We can use equation (30) to eliminate $p_+$ from equation (31) giving,

$$p_+ = 1 - \lambda - p_- \quad \Rightarrow \quad \mu = \lambda + p_- b_- + (1 - \lambda - p_-)b_+$$
$$= b_+ - (b_+ - 1)\lambda - (b_+ - b_-)p_-$$

Rearranging allows us to write $p_-$ as a function of $\mu$ and $\lambda$,

$$\Rightarrow \quad p_- = \frac{b_+ - \mu - (b_+ - 1)\lambda}{b_+ - b_-} \tag{34}$$

We can now eliminate both $p_\pm$ from equation (29), starting with $p_+$,

$$\mu^\alpha I(\boldsymbol{b}; \alpha) = (1 - \lambda)f_\alpha(b_+) - p_-[f_\alpha(b_+) - f_\alpha(b_-)] - f_\alpha(\mu).$$

Substituting equation (34) to eliminate $p_-$ gives,

$$\mu^\alpha I(\boldsymbol{b}; \alpha) = (1 - \lambda)f_\alpha(b_+) - [b_+ - \mu - (b_+ - 1)\lambda]\beta_\alpha - f_\alpha(\mu),$$

where $\beta_\alpha = [f_\alpha(b_+) - f_\alpha(b_-)]/(b_+ - b_-)$. Grouping terms in $\lambda$, and rearranging gives,

$$I(\boldsymbol{b}; \alpha) = [(b_+ - \mu)(r_\alpha(\mu, b_+) - \beta_\alpha) - (b_+ - 1)(r_\alpha(1, b_+) - \beta_\alpha)\lambda]/\mu^\alpha. \tag{35}$$

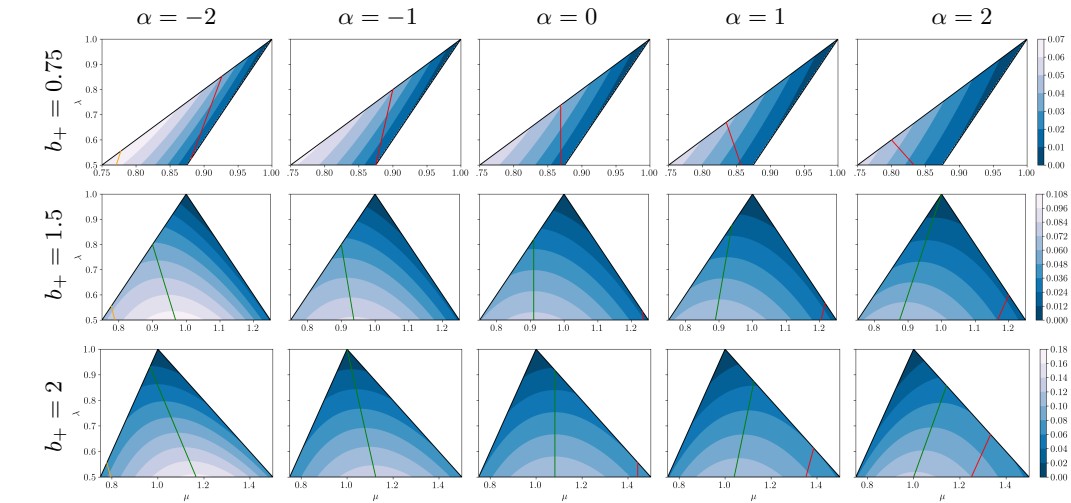

Figure 4: Index surface in $(\mu, \lambda)$ space for varying benefit functions and generalization parameter values. Here we hold $b_- = 0.5$ constant and vary $b_+$ and $\alpha$.

Since $f_\alpha$ is convex, we know that $r_\alpha$ is strictly increasing. Looking at equation (35), since both $\mu$ and unity are both greater than $b_-$, all the terms in square parenthesis are positive; as is the denominator $\mu^\alpha$. Thus we can see that $I(\boldsymbol{b}; \alpha)$ is linear with respect to $\lambda$. The index is a linearly decreasing function of accuracy for $b_+ > 1$ and increasing for $b_+ < 1$.

$$\mu^\alpha I(\boldsymbol{b}; \alpha) = f_\alpha(b_+) - f_\alpha(\mu) - (b_+ - \mu)\beta_\alpha - [f_\alpha(b_+) - (b_+ - 1)\beta_\alpha]\lambda$$
$$= \beta_\alpha \mu - f_\alpha(\mu) + f_\alpha(b_+) - b_+ \beta_\alpha - [f_\alpha(b_+) - b_+ \beta_\alpha + \beta_\alpha]\lambda$$
$$= A_\alpha + \beta_\alpha \mu - (A_\alpha + \beta_\alpha)\lambda - f_\alpha(\mu)$$

In Fig. 4, we plot the index surface as a contour plot for a variety of parameter choices. □

**Corollary 3.2.1** (Behavior with respect to unit reward rate) *For benefits $b_i \in \{b_-, b_+, 1\}$ and fixed $\mu$, the index is a linear function of the unit reward rate, for $b_+ < 1$, it is increasing and for $b_+ > 1$, it is decreasing. When $b_+ = 1$, the index is independent of the unit reward rate.*

*Proof.* Figure 2 illustrates the behavior of $f_\alpha(x)$ around $x = 1$, where $f_\alpha(1) = 0$. We can write the equation of the secant line passing through $f_\alpha(b_-)$ and $f_\alpha(b_+)$ as $y = A_\alpha + \beta_\alpha x$. Thus, $A_\alpha + \beta_\alpha$ is the value on this line at $x = 1$. Since $f_\alpha(1) = 0$, and $f_\alpha(x)$ is convex, we know that $A_\alpha + \beta_\alpha$ is negative for $b_+ < 1$ and positive for $b_+ > 1$. Importantly $A_\alpha + \beta_\alpha = 0$ only if one of $b_- = 1$ or $b_+ = 1$. □

**Corollary 3.2.2** (Behavior with respect to the mean benefit). *For benefits $b_i \in \{b_-, b_+, 1\}$ and fixed $\lambda$, the index has a single turning point at $\mu = \hat{\mu}(\lambda)$, where,*

$$\hat{\mu}(\lambda) = \begin{cases} -1/\beta_0 & \text{for} \quad \alpha = 0 \\ (A_1 + \beta_1)\lambda - A_1 & \text{for} \quad \alpha = 1 \\ \dfrac{\alpha(\alpha - 1)[(A_\alpha + \beta_\alpha)\lambda - A_\alpha] - 1}{(\alpha - 1)^2 \beta_\alpha} & \text{otherwise.} \end{cases} \tag{7}$$

*In most cases the turning point is a maximum turning point. In the special case where $b_- = 0$, as we increase $b_+$, the turning point changes from a minima (for $b_+ < 1$) to an inflection point (at $b_+ = 1$), and finally a maxima (for $b_+ > 1$).*

*Proof.* Proof here. □

**Theorem 3.5** (Index as a function of the error distribution for $\lambda = \hat{p}$). *For the benefit function* $b_{ij} = ((b_+, b_-), (1, 1))$, *the index $I(\boldsymbol{b}; \alpha)$ can be written as a function of the false positive (FPR), negative (FNR) and data and model reward rates $p = \mathbb{P}(Y = 1)$ and $\hat{p} = \mathbb{P}(\hat{Y} = 1)$*

$$I(\boldsymbol{b}; \alpha) = \left[ pFNRf_\alpha(b_-) + q(1 - FPR)f_\alpha(b_+) - f_\alpha(\mu) \right]/\mu^\alpha$$
$$\mu = 1 - (1 - b_+)q(1 - FPR) - (1 - b_-)pFNR.$$

*Proof.* Eqs. (29) and (32) still hold true but now $p_- = pFNR$ and $p_+ = q(1 - FPR)$. Substituting completes the proof. ☐

**Theorem 3.4** (Index as a function of the error distribution for $\lambda = 1 - \hat{p}$). *For the benefit function* $b_{ij} = ((1, 1), (b_+, b_-))$, *the index $I(\boldsymbol{b}; \alpha)$ can be written as a function of the false positive (FPR), negative (FNR) and data and model reward rates $p = \mathbb{P}(Y = 1)$ and $\hat{p} = \mathbb{P}(\hat{Y} = 1)$*

$$I(\boldsymbol{b}; \alpha) = \left[ p(1 - FNR)f_\alpha(b_-) + qFPRf_\alpha(b_+) - f_\alpha(\mu) \right]/\mu^\alpha$$
$$\mu = 1 - (1 - b_-)p(1 - FNR) - (1 - b_+)qFPR.$$

*Proof.* Eqs. (29) and (32) still hold true but now $p_- = p(1 - FNR)$ and $p_+ = qFPR$. Substituting completes the proof. ☐

**The subset of benefit functions and generalization parameters which result in a metric which can be used to satisfy error distribution bounds pre-training** . *For the benefits in $\{\varphi b, b, 1\}$ and values of $\alpha \in (0, 1)$ the index is a monotonic function of the mean benefit $\mu$, provided ensures the the probability of an undesirable error is at most $\kappa$ times the probability of a benign error.*

*Proof.* Recall, the index maxima location is given by Eq. (7),

$$\hat{\mu}(\lambda) = \begin{cases} -1/\beta_0 & \text{for} \quad \alpha = 0 \\ (A_1 + \beta_1)\lambda - A_1 & \text{for} \quad \alpha = 1 \\ \dfrac{\alpha(\alpha - 1)[(A_\alpha + \beta_\alpha)\lambda - A_\alpha] - 1}{(\alpha - 1)^2 \beta_\alpha} & \text{otherwise.} \end{cases} \tag{7}$$

Let's consider the behavior of the maxima $\hat{\mu}(\lambda)$ for different possible values of $\alpha$, starting with the simplest,

$$\alpha = 0 \ \Rightarrow \ (5) \ \Rightarrow \ \beta_0 = \frac{-\ln \varphi}{(\varphi - 1)b}, \quad (7) \ \Rightarrow \ \hat{\mu} = \frac{-1}{\beta_0} = \frac{\varphi - 1}{\ln \varphi}b \tag{36}$$

$$\alpha = 1 \ \Rightarrow \ (5) \ \Rightarrow \ \beta_1 = \ln b + \frac{\varphi \ln \varphi}{\varphi - 1}, \quad A_1 = -\frac{\varphi \ln \varphi}{\varphi - 1}b$$

$$\Rightarrow \ (7) \ \Rightarrow \ \hat{\mu}(\lambda) = (A_1 + \beta_1)\lambda - A_1 = \frac{\varphi \ln \varphi}{\varphi - 1}[b + (1 - b)\lambda] + \lambda \ln b \tag{37}$$

$$\alpha \notin \{0, 1\} \ \Rightarrow \ (5) \ \Rightarrow \ \beta_\alpha = \frac{(\varphi^\alpha - 1)b^{\alpha-1}}{\alpha(\alpha - 1)(\varphi - 1)}, \quad A_\alpha = \frac{(b^\alpha - 1)(\varphi - 1) - (\varphi^\alpha - 1)b^\alpha}{\alpha(\alpha - 1)(\varphi - 1)}$$

$$\Rightarrow \ \frac{A_\alpha}{\beta_\alpha} = \frac{(\varphi - 1)(b - b^{1-\alpha})}{\varphi^\alpha - 1} - b, \quad \frac{A_\alpha}{\beta_\alpha} + \frac{1}{\alpha(\alpha - 1)\beta_\alpha} = \left( \frac{\varphi - 1}{\varphi^\alpha - 1} - 1 \right)b$$

$$\Rightarrow \ (7) \ \Rightarrow \ \hat{\mu}(\lambda) = \frac{\alpha}{\alpha - 1}\left[ \left( \frac{A_\alpha}{\beta_\alpha} + 1 \right)\lambda - \left( \frac{A_\alpha}{\beta_\alpha} + \frac{1}{\alpha(\alpha - 1)\beta_\alpha} \right) \right]$$

$$\Rightarrow \ \hat{\mu}(\lambda) = \frac{\alpha}{\alpha - 1}\left[ \left( \frac{\varphi - 1}{\varphi^\alpha - 1}(b - b^{1-\alpha}) - b + 1 \right)\lambda + \left( 1 - \frac{\varphi - 1}{\varphi^\alpha - 1} \right)b \right]$$

$$\Rightarrow \ \hat{\mu}(\lambda) = \frac{\alpha[(b^{1-\alpha} - \hat{\varphi}b^{1-\alpha} - b + \hat{\varphi})b)\lambda + (1 - \hat{\varphi})b]}{\alpha - 1} \quad \text{where} \quad \hat{\varphi} = \frac{1 - \varphi}{1 - \varphi^\alpha}. \tag{38}$$

Putting together Eqs. (36)-(38),

$$
\hat{\mu}(\lambda) = \begin{cases}
\dfrac{\varphi - 1}{\ln \varphi} b & \text{for} \quad \alpha = 0 \\[2mm]
\dfrac{\varphi \ln \varphi}{\varphi - 1}[b + (1 - b)\lambda] + \lambda \ln b & \text{for} \quad \alpha = 1 \\[2mm]
\dfrac{\alpha[b + (1 - b)\lambda]}{\alpha - 1} - \dfrac{\alpha \hat{\varphi} b[1 + (b^{-\alpha} - 1)\lambda]}{\alpha - 1} & \text{otherwise.}
\end{cases}
$$

**Avoiding harm**   For the benefit matrix $b_{ij} = ((1,1),(\varphi b, b))$. The index is strictly increasing in $\mu$ if and only if $\hat{\mu}(\lambda) > b + (1 - b)\lambda$.

$$\Leftrightarrow \qquad \hat{\mu}(\lambda) - [b + (1 - b)\lambda] > 0$$

$$
\Leftrightarrow \quad \begin{cases}
\left(\dfrac{\varphi - 1}{\ln \varphi} - 1\right) b - (1 - b)\lambda & \text{for} \quad \alpha = 0 \\[2mm]
\left(\dfrac{\varphi \ln \varphi}{\varphi - 1} - 1\right)[b + (1 - b)\lambda] + \lambda \ln b & \text{for} \quad \alpha = 1 \\[2mm]
\dfrac{[b + (1 - b)\lambda] - \alpha \hat{\varphi} b[1 + (b^{-\alpha} - 1)\lambda]}{\alpha - 1} & \text{otherwise.}
\end{cases} \quad \Biggr\} > 0
$$

$$b = 1 - \varepsilon \quad \text{and} \quad \varphi = 1 - \epsilon < 1 \quad \text{and} \quad \ln \varphi = \ln(1 - \epsilon) = -\epsilon\left(1 + \frac{\epsilon}{2} + \mathrm{O}(\epsilon^2)\right)$$

$$\Rightarrow \quad \frac{\varphi - 1}{\ln \varphi} = \left(1 + \frac{\epsilon}{2} + \mathrm{O}(\epsilon^2)\right)^{-1} = 1 - \frac{\epsilon}{2} + \mathrm{O}(\epsilon^2) < 1$$

$$\Rightarrow \quad \frac{\varphi \ln \varphi}{\varphi - 1} = (1 - \epsilon)\left(1 + \frac{\epsilon}{2} + \mathrm{O}(\epsilon^2)\right) = 1 - \frac{\epsilon}{2} + \mathrm{O}(\epsilon^2) < 1$$

$$\Rightarrow \quad \hat{\varphi} = (1 - \varphi)(1 - \varphi^\alpha)^{-1} = \epsilon[1 - (1 - \epsilon)^\alpha]^{-1}$$

$$\Rightarrow \quad \hat{\varphi} = \epsilon\left[1 - \left(1 - \alpha\epsilon + \frac{\alpha(\alpha - 1)\epsilon^2}{2} + \mathrm{O}(\epsilon^3)\right)\right]^{-1} = \epsilon\left(\alpha\epsilon - \frac{\alpha(\alpha - 1)\epsilon^2}{2} + \mathrm{O}(\epsilon^3)\right)^{-1}$$

$$\Rightarrow \quad \alpha\hat{\varphi} = 1 + \frac{(\alpha - 1)\epsilon}{2} + \mathrm{O}(\epsilon^2) \quad \Rightarrow \quad \alpha\hat{\varphi} - 1 = \frac{(\alpha - 1)\epsilon}{2} + \mathrm{O}(\epsilon^2).$$

For $\alpha > 1$ we need,

$$\alpha\hat{\varphi}b[1 + (b^{-\alpha} - 1)\lambda] < [b + (1 - b)\lambda]$$

$$\Leftrightarrow \quad (\alpha\hat{\varphi} - 1)b < [1 - b + \alpha\hat{\varphi}b(1 - b^{-\alpha}))]\lambda$$

$$\Leftrightarrow \quad (\alpha\hat{\varphi} - 1)b < [(\alpha\hat{\varphi} - 1)b - (\alpha\hat{\varphi}b^{1-\alpha} - 1)]\lambda$$

$$b < 1 \quad \Rightarrow \quad b^{1-\alpha} > 1 \quad \Rightarrow \quad 0 < \alpha\hat{\varphi} - 1 < \alpha\hat{\varphi}b^{1-\alpha} - 1$$

$$\Leftrightarrow \quad b < \left(b - \frac{\alpha\hat{\varphi}b^{1-\alpha} - 1}{\alpha\hat{\varphi} - 1}\right)\lambda < (b - 1)\lambda < 0$$

For $\alpha < 1$ we need,

$$(\alpha\hat{\varphi} - 1)b > [(\alpha\hat{\varphi} - 1)b - (\alpha\hat{\varphi}b^{1-\alpha} - 1)]\lambda$$

$$b < 1 \quad \Rightarrow \quad b^{1-\alpha} < 1 \quad \Rightarrow \quad \alpha\hat{\varphi} - 1 < \alpha\hat{\varphi}b^{1-\alpha} - 1 < 0$$

$$\Leftrightarrow \quad 0 < b < \left(b - \frac{\alpha\hat{\varphi}b^{1-\alpha} - 1}{\alpha\hat{\varphi} - 1}\right)\lambda$$

$$\Leftrightarrow \quad b > \frac{\alpha\hat{\varphi}b^{1-\alpha} - 1}{\alpha\hat{\varphi} - 1} = (1 - \alpha\hat{\varphi}b^{1-\alpha})(1 - \alpha\hat{\varphi})^{-1}$$

$$= \left[1 - \left(1 - \frac{(1 - \alpha)\epsilon}{2} + \mathrm{O}(\epsilon^2)\right)b^{1-\alpha}\right]\left(\frac{(1 - \alpha)\epsilon}{2} + \mathrm{O}(\epsilon^2)\right)^{-1}$$

$$\Leftrightarrow \quad b > \frac{2}{(1 - \alpha)\epsilon}\left(1 - b^{1-\alpha} + \frac{(1 - \alpha)\epsilon}{2}b^{1-\alpha}\right) + \mathrm{O}(\epsilon)$$

$$\Leftrightarrow \quad b > \frac{2(1 - b^{1-\alpha})}{(1 - \alpha)\epsilon} + b^{1-\alpha} + \mathrm{O}(\epsilon) \quad \text{where} \quad \epsilon = \frac{(\tilde{\kappa} - 1)(1 - b)}{b}$$

$$\Leftrightarrow \quad b > \frac{2b(1 - b^{1-\alpha})}{(\tilde{\kappa} - 1)(1 - \alpha)(1 - b)} + b^{1-\alpha} + \mathrm{O}(\epsilon)$$

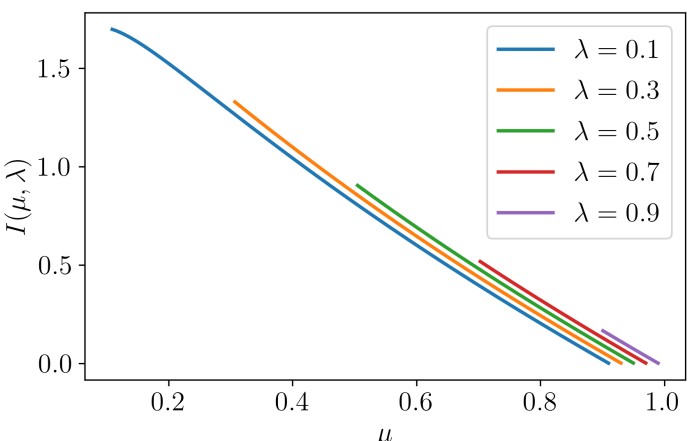

Figure 5: $FPR$ against $FNR$ when $b_{ij} = [[0.9, 0.01], [1, 1]]$ and $\alpha = 0.5$.

**Avoiding undue credit** For the benefit matrix, $b_{ij} = ((\varphi b, b), (1, 1))$, the index is strictly decreasing in $\mu \Leftrightarrow \hat{\mu}(\lambda) < \varphi b + (1 - \varphi b)\lambda$. Equality holds when,

$$\Leftrightarrow \quad \hat{\mu}(\lambda) - [\varphi b + (1 - \varphi b)\lambda] = 0$$

$$\Leftrightarrow \quad \begin{cases} \left(\dfrac{\varphi - 1}{\ln \varphi} - 1\right) b - (1 - b)\lambda & \text{for} \quad \alpha = 0 \\[2ex] \left(\dfrac{\varphi \ln \varphi}{\varphi - 1} - 1\right) [b + (1 - b)\lambda] + \lambda \ln b & \text{for} \quad \alpha = 1 \\[2ex] \dfrac{[b + (1 - b)\lambda] - \alpha \hat{\varphi} b[1 + (b^{-\alpha} - 1)\lambda]}{\alpha - 1} & \text{otherwise.} \end{cases} = 0$$

$\square$

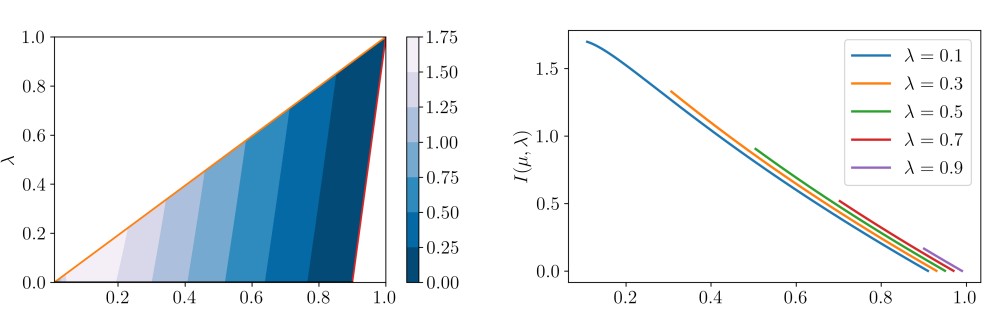

Figure 6: Birds-eye view (left) and side view (right) of the index surface when $\alpha = 0.5$ and $b_{ij} = [[0.9, 0.01], [1, 1]]$.

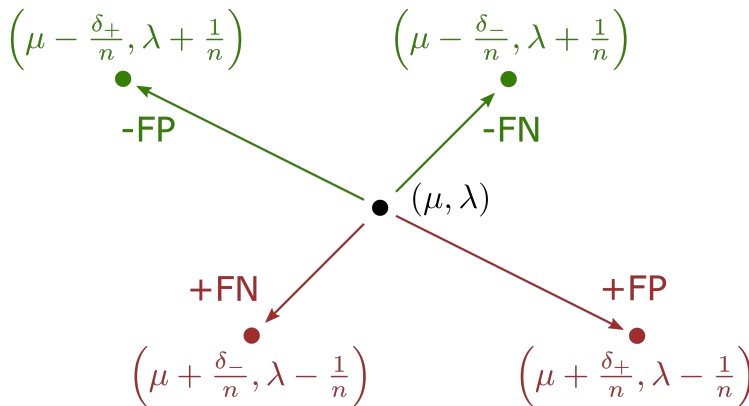

Figure 7: Finite difference grid showing those models neighbouring that at $(\mu, \lambda)$ given $n$, when $(\mu, \lambda)$ is not on an edge.

**The Deviation Region**    In this section we will identify the *deviation region*, that is, the part of model performance space for which fairness and accuracy are opposed. To do this, we need to know the cost of an error as a function of the mean benefit $\mu$ and the model accuracy, $\lambda$. Fairness and accuracy are opposed when the cost of an error becomes negative. Let us denote the *cost of an error* as,

$$\Delta I^{\pm}(\boldsymbol{b}; \alpha) = I(\boldsymbol{b}^{\pm}) - I(\boldsymbol{b}; \alpha). \tag{39}$$

Here $\boldsymbol{b}^{\pm}$ differs from $\boldsymbol{b}$ by one prediction only, containing one less correct prediction, and one more erroneous one. An additional error decreases the accuracy by $1/n$ and changes the mean benefit by $\delta_{\pm}/n$, where $\delta^{\pm} = b_{\pm} - 1$. Since $b_- < 1$, $\delta_- < 0$; while $\delta_+$ may be positive or negative. The discrete grid of points that we can reach through a small change in model performance on a set of $n$ individuals (given $\mu, \lambda$), is shown in Figure 7. Again, for illustration purposes only, we assume $b_+ > 1$. Together Figures 1 and 7 provide a global and local view of the model performance space which is traversed during model training. The bottom left corner of the triangle is the model for which all errors are false negatives, that is the algorithm rewards no one (or harms everyone), and the bottom right corner is the model which rewards everyone (or harms no one), assuming the $p = 50\%$. At the top of the triangle is the oracle, a model which is able to perfectly separate positive and negative classes in the training data. If we apply a threshold on the proportion of individuals who are rewarded, as we increase the threshold from zero to one, the oracle traverses the top edges of the triangle, from the bottom left corner, to the top and down the right edge. For any given model, making one additional error moves us downwards and parallel to the left or right edge of the triangle, depending on whether the error is a false negative of false positive respectively.

Using this we can calculate the cost of different errors as a function of $\mu$ and $\lambda$.

**Theorem C.4** (The Cost of Errors). *For benefits $b_{ij} = ((1, b_-), (b_+, 1))$ and large $n$, the cost of an error can be written as*

$$\Delta I^{\pm}(\mu, \lambda) = \xi_{\alpha}^{\pm}(\mu, \lambda)/n + \mathrm{O}(1/n^2)$$

*where,*

$$\xi_{\alpha}^{\pm}(\mu, \lambda) = (b_{\pm} - 1)(C_{\mu}^{\pm}\mu + C_{\lambda}\lambda - C_0)/\mu^{\alpha+1}, \tag{40}$$

*and*

$$\left. \begin{array}{l} C_{\mu}^{\pm} = r_{\alpha}(1, b_{\pm}) - \alpha\beta_{\alpha} + \mathbb{1}(\alpha - 1), \\ C_{\lambda} = \alpha(A_{\alpha} + \beta_{\alpha}), \\ C_0 = \alpha A_{\alpha} + [1 - \mathbb{1}(\alpha - 1)]/(\alpha - 1). \end{array} \right\} \tag{41}$$

*$A_{\alpha}$ and $\beta_{\alpha}$ are defined in equation (5) and $\mathbb{1}(x) = 1$ if $x = 0$ and zero otherwise. For $\alpha = 0$, and $b_{\pm} = 1$, $C_{\lambda} = 0$ making the cost of an error independent of the unit reward rate $\lambda$. We can write equation (40) as,*

$$\xi_{\alpha}^{\pm}(\mu, \lambda) = (b_{\pm} - 1)C_{\mu}^{\pm}[\mu - \mu_{\pm}^*(\lambda)] \tag{42}$$

*where,*

$$\mu_{\pm}^*(\lambda) = (C_0 - C_{\lambda}\lambda)/C_{\mu}^{\pm}. \tag{43}$$

*Thus, the cost of an error is zero for $b_\pm = 1$, $C_\mu^\pm = 0$, and when $\mu = \mu_\pm^*(\lambda)$. See appendix C.2 for the proof.*

*Proof.* Eqs. (29) and (32) provide an expression for $I(\boldsymbol{b}; \alpha)$.

$$I(\boldsymbol{b}; \alpha) = [f_\alpha(b_-)p_- + f_\alpha(b_+)p_+ - f_\alpha(\mu)]/\mu^\alpha$$
$$\text{where} \quad \mu = 1 + (b_- - 1)p_- + (b_+ - 1)p_+$$

$\boldsymbol{b}^\pm$ differs from $\boldsymbol{b}$ by one prediction only, containing one less correct prediction, and one more erroneous (either a false positive or negative) one. An additional error decreases the accuracy by $1/n$ and changes the mean benefit by $\delta_\pm/n$, where $\delta_\pm = b_\pm - 1$. Thus,

$$I(\boldsymbol{b}^\pm) = \left[ f_\alpha(b_-)p_- + f_\alpha(b_+)p_+ + \frac{f_\alpha(b_\pm)}{n} \right.$$
$$\left. - f_\alpha\left(\mu + \frac{\delta_\pm}{n}\right) \right] \left(1 + \frac{\delta_\pm}{n\mu}\right)^{-\alpha} \frac{1}{\mu^\alpha}.$$

$$\text{Since} \quad f_\alpha\left(\mu + \frac{\delta_\pm}{n}\right) = f_\alpha(\mu) + f_\alpha'(\mu)\left(\frac{\delta_\pm}{n}\right) + O\left[\left(\frac{\delta_\pm}{n}\right)^2\right]$$

$$\text{and} \quad \left(1 + \frac{\delta_\pm}{n\mu}\right)^{-\alpha} = 1 - \frac{\delta_\pm \alpha}{n\mu} + O\left[\left(\frac{\delta_\pm}{n\mu}\right)^2\right]$$

$$\Rightarrow \quad I(\boldsymbol{b}^\pm) = \left[ I(\boldsymbol{b}; \alpha) + \frac{f_\alpha(b_\pm) - \delta_\pm f_\alpha'(\mu)}{n\mu^\alpha} \right] \left(1 - \frac{\delta_\pm \alpha}{n\mu}\right) + O\left[\left(\frac{1}{n}\right)^2\right],$$

$$= I(\boldsymbol{b}; \alpha) + \frac{f_\alpha(b_\pm) - \delta_\pm f_\alpha'(\mu)}{n\mu^\alpha} - \frac{\delta_\pm \alpha I(\boldsymbol{b}; \alpha)}{n\mu} + O\left[\left(\frac{1}{n}\right)^2\right].$$

For large $n$, we can write the cost of an error as

$$\Delta I^\pm(\mu, \lambda; n) = \xi_\alpha^\pm(\mu, \lambda)/n + O(1/n^2)$$

where,

$$\xi_\alpha^\pm(\mu, \lambda) = [[f_\alpha(b_\pm) - \delta_\pm f_\alpha'(\mu)]\mu - \delta_\pm \alpha \mu^\alpha I(\boldsymbol{b}; \alpha)]/\mu^{\alpha+1}.$$

We know that, $f_\alpha(b_\pm)/\delta_\pm = r_\alpha(1, b_\pm)$, thus

$$\xi_\alpha^\pm(\mu, \lambda) = \delta_\pm [[r_\alpha(1, b_\pm) - f_\alpha'(\mu)]\mu - \alpha \mu^\alpha I(\boldsymbol{b}; \alpha)]/\mu^{\alpha+1}.$$

Substituting equation (4) for $I(\boldsymbol{b}; \alpha)$

$$\xi_\alpha^\pm(\mu, \lambda) = \delta_\pm[[r_\alpha(1, b_\pm) - f_\alpha'(\mu)]\mu$$
$$- \alpha[A_\alpha(1 - \lambda) + \beta_\alpha(\mu - \lambda) - f_\alpha(\mu)]]/\mu^{\alpha+1}$$
$$= \delta_\pm[[r_\alpha(1, b_\pm) - \alpha\beta_\alpha]\mu - [f_\alpha'(\mu)\mu - \alpha f_\alpha(\mu)]$$
$$- \alpha[A_\alpha - (A_\alpha + \beta_\alpha)\lambda]]/\mu^{\alpha+1}$$

where $A_\alpha$ and $\beta_\alpha$ are defined in equation (5). Using equation (20) we can show,

$$f_\alpha'(\mu)\mu - \alpha f_\alpha(\mu) = \begin{cases} \mu & \text{for } \alpha = 1 \\ 1/(\alpha - 1) & \text{otherwise.} \end{cases}$$

Substituting gives,

$$\alpha = 1 \Rightarrow \xi_\alpha^\pm(\mu, \lambda) = \delta_\pm[[r_1(1, b_\pm) - (\beta_1 + 1)]\mu$$
$$- [A_1 - (A_1 + \beta_1)\lambda]]/\mu^2,$$
$$\alpha \neq 1 \Rightarrow \xi_\alpha^\pm(\mu, \lambda) = \delta_\pm[[r_\alpha(1, b_\pm) - \alpha\beta_\alpha]\mu$$
$$- [1/(\alpha - 1) + \alpha A_\alpha - \alpha(A_\alpha + \beta_\alpha)\lambda]]/\mu^{\alpha+1}$$

where $\delta_\pm = b_\pm - 1$. Therefore we can write,

$$\xi_\alpha^\pm(\mu, \lambda) = (b_\pm - 1)(C_\mu^\pm \mu + C_\lambda \lambda - C_0)/\mu^{\alpha+1}. \tag{40}$$

$\square$

# D    EMPIRICAL EVIDENCE

## D.1    ADULT DATASET

**Model Performance Metrics**    In Fig. 8, we plot the accuracy and error rates on the left plot and their differences on the right.

**Comparing Indices with Model Performance Metrics**

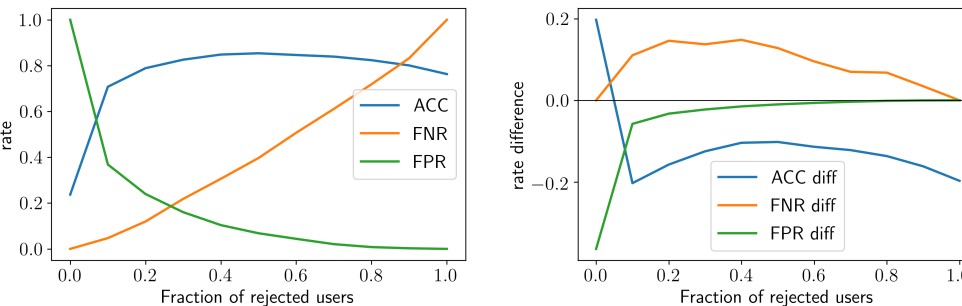

Figure 8: .

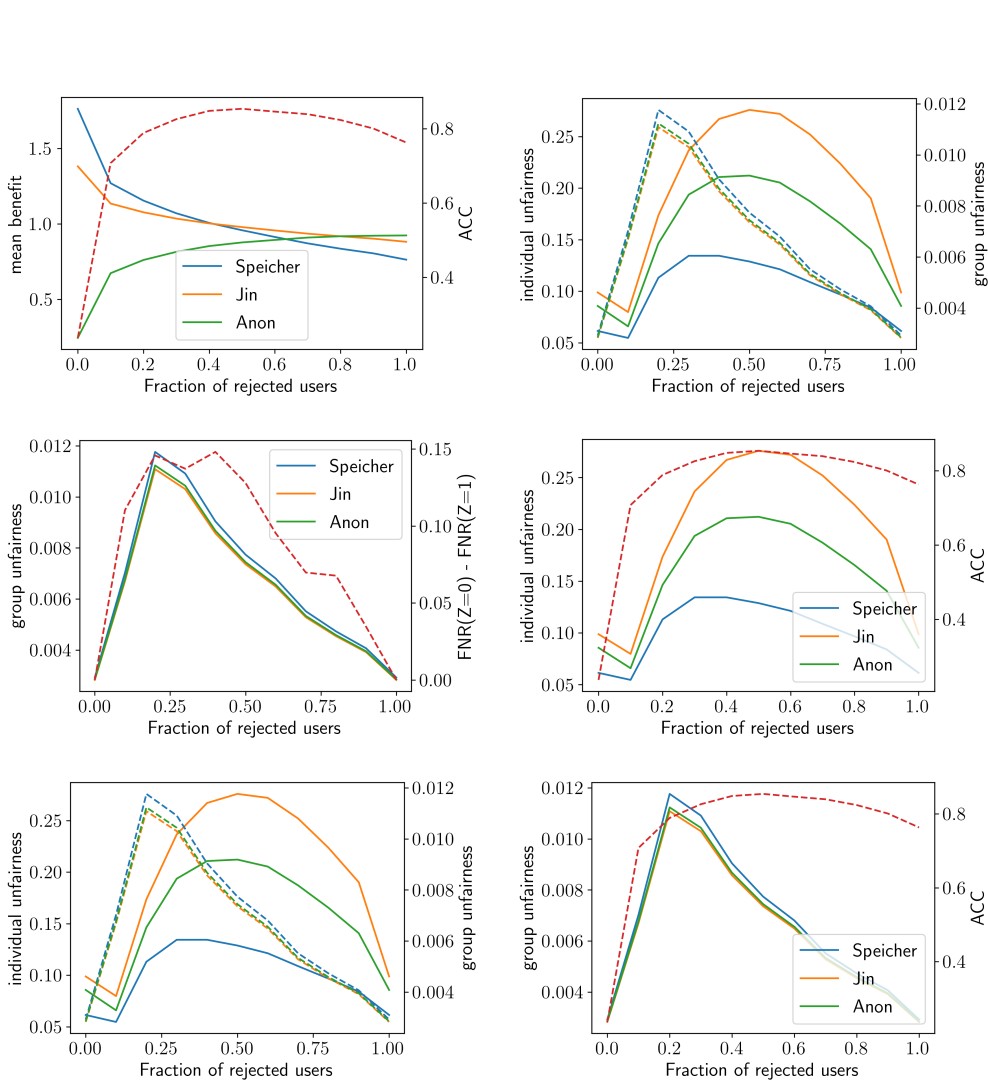

Figure 9: .

