# OpenReview forum: "Utility as Fair Pricing"
_ICLR.cc/2025/Conference — Submitted to ICLR 2025_

### Official Review · Reviewer_cEeF · 2024-10-21

**Soundness:** 2
**Presentation:** 2
**Contribution:** 2
**Rating:** 3
**Confidence:** 3

**Summary:**

The primary goal of the work is to obtain a deeper understanding of the class of generalized entropy index based unfairness metrics proposed in [1]; the primary tool the authors use to achieve this is deriving more interpretable representations of the aforementioned unfairness metrics. The motivation for this process is to allow for better parameter selection in generalized entropy indices which are used to produce fair models. To support this, the authors use their theoretical contributions to provide a characterization parameters that allow for enforcement of individual fairness. Theoretical examples are also given.

[1] Speicher et al. A Unified Approach to Quantifying Algorithmic Unfairness: Measuring Individual & Group Unfairness via Inequality Indices. 2018.

**Strengths:**

1. The authors do a good job explaining that while generalized entropy indices for fairness is common in practice, the parameter selection methods for these metrics are ad-hoc and often not well justified.
2. I appreciate the avoiding harm and avoiding undue credit as theoretical examples of how Theorem 3.3 and Theorem 3.4 can aid in a better parameter selection process.
3. Similarly, I can see how the characterization given in Theorem 3.5 would be useful for fairness parameter selection.

**Weaknesses:**

1. The primary weakness of the work is the lack of any empirical evidence that demonstrate the usefulness of the results. While the authors take care to derive several new representations for the generalized entropy indices and provide some theoretical examples for how these representations can help with parameter selection for fairness criteria, basically no empirical evidence is provided. In particular, I believe that the work requires at least a couple experiments that (i): show that naive or standard parameter selection leads to poor fairness performance and (ii) Show that selection of parameters suggested by the authors theory alleviates this issue. It seems the major works theirs is based off all include experiments with real data, so I believe this is a reasonable ask.

2. In general, the writing needs work. I provide further suggestions on this below.

*More detailed writing comments*
1. I think the primary objective/motivation is not super clear. My interpretation is the point is to "aid with fairness parameter selection", but I don't think this is demonstrated well enough. One suggestion I have is to re-organize the main contributions list in the introduction; I think it is sufficient to pick the 1-2 broad points the reader should take away from the paper, rather than list every idea in the work.
2. Please provide a formal, mathematical definition in your notation for the risk free reward $\lambda$ (I understand that this can change based on choices for the benefit function, maybe this can be rolled in with point 4).
3. Line 364; please provide a further discussion on how each corollary demonstrates the observation that a poor choice of parameters leads to a metric that is behaving opposite to which it should. This observation is not immediate it to me just examining the corollaries.
4. Around line 220; I think it would be very helpful to have a plot or table demonstrating the relationship between the various choices to assign to $\lambda$, and the benefit map $B$. Similar to this, when a choice for $b_{ij}$ is made (for example line 378) it should be more clear what position in the Matrix corresponds to what instance (e.g false positive, false negative etc...). In general I found this hard to track throughout the paper
5. Line 449 has an incomplete sentence
6. The conclusion/discussion section has multiple ideas that require clarification. For example, the final bit " In some sense, the choice α = 0 results in a mis-pricing of an individual not dissimilar to the mis-pricing of vanilla options under the assumption of constant volatility demonstrated by Black Scholes and Merton’s log-normal model which empirically demonstrates the existence of volatility smile. Here the invalid assumption at the root of the mis-pricing, is that errors and group membership are uncorrelated." is an interesting analogy for the phenomenon being discussed, but I do not think that readers at an ML conference are familiar enough with Merton's log-normal model for this to be particularly meaningful. Moreover, references to these models are provided here. I felt this was a repeating pattern in this section.

**Questions:**

1. The authors state that one primary contribution is to "argue that in order to represent individual fairness, the index must be orthogonal to model accuracy. For the parameter choices made Speicher et al. (2018), we show that the index is a linear function of model accuracy, and thus cannot represent individual fairness according to this independence constraint..." However, I can not find where exactly this is discussed in the main text. This appears to be a different finding/interpretation from the Speicher et al. paper, so it warrants thorough discussion. Is there somewhere I am missing where this occurs?
2. Theorem 3.5 provides a parameter selection characterization for individual fairness. Is there somewhere in the literature that has an analogous result for group fairness? If not, is this one potential follow up direction that could be pursued?

---

> ### Author Response · Authors · 2024-11-13
> **cEeF comments and questions**
>
> - Line 364: Corollary 3.2.1 says that the index could be either decreasing of increasing in $\lambda$ depending on the parameter choices. Corollary 3.2.2. says that the index could have a maxima or minima in $\mu$. The turning point may or may not fall within the index domain described in equation (6) for a given value of $\lambda$. If the turning point falls outside of the domain then the index is monotonic in $\mu$. Thus, whatever relation we might expect/want the index to have with respect to $\lambda$ or $\mu$, it is possible for the index to display the opposite (increasing instead of decreasing, maxima instead of minima, for example) by choosing index parameters ($b_{ij}$ and $\alpha$) accordingly.
>
> - Line 449: We could not find the incomplete sentence you mention. If you could paste the line, that would be helpful.
>
> - Q1A: All prior works in Table 1 assume that accurate predictions are equally beneficial, that is, $b_{ij} =\mathrm{benefit}(\hat{y}=i, y=j) = ((1, b_{FN}),(b_{FP}, 1))$ where $b_{FN}$ and $b_{FP}$ are the false positive and negative benefits. All authors in Table 1 effectively assume that the *unit reward rate* (proportion of individuals receiving the unit reward) is the model accuracy (since only accurate predictions are awarded the unit benefit). Theorem 3.2 then shows that the index is a function of the unit reward rate (model accuracy in prior works) and mean benefit. This does not contradict the findings in Speicher et al. The index is linear in model accuracy for *fixed* mean benefit. Proposition 3.2 in Speicher et al. says the for non-perfect classifiers, the fairness and accuracy optimal classifiers do not coincide. This is true because in constructing one, they do not hold the mean benefit constant. Moreover, the classifier they construct to prove this proposition is not viable because it has an accuracy of less than 0.5.
>
> We will address the remaining issues in a separate response.

---

> > ### Comment · Reviewer_cEeF · 2024-11-19
> >
> > Re line 449, the sentences are technically complete but the phrasing is choppy, I suggest adjusting the wording.
> >
> > Thank you for addressing my questions on corollary 3.2.1 and the findings in Speicher et al. I think this work has potential and the core ideas on improving parameter selection for GEI are interesting, that said, I still believe the work is greatly missing empirical evidence. I understand that your contour plots are a good verification of the theory, but proofs already provide this, and these are not a substitution for demonstrations of the theory in practice (e.g. using the characterizations to select fairness parameters on a real data set). The prior work on GEI the authors build off includes this, and one of the stated motivations of the paper is the many open source libraries where GEI's are implemented, given this, I do not see the major obstacle to including something in this vein. As other reviewers noted, the paper is quite dense and has a long discussion section, I also believe that including concrete examples will improve the readability (and impact) of the work a great deal.

---

> ### Author Response · Authors · 2024-11-16
> **cEeF: Final remarks and questions**
>
> Thank you for your helpful suggestions, we agree, the introduction, contributions and discussion could be improved substantially. We will remove the reference to the deviation region which is out of place. There are missing references in the discussion, which was rushed; we can certainly provide these. We wonder if discussions on the connection with derivatives hinders more than it helps? Employment is a tangible example with obvious financial value attached, and is referenced in the introduction, so makes sense. There is also the recidivism risk example in Blackstone's formulation but this is clearly a harder problem to engage with than employment, since many more people have to get jobs, than deal with the criminal justice system. Additionally, when the latter happens, participation is usually involuntary and the stakes are higher.
>
> Q2A. We argue that a different choices of $\alpha$ changes the relative prioritization between-group and within-group fairness. We expect that this insight can be used to mitigate between-group unfairness, by discounting the within-group component for wealthier groups. We can certainly discuss this more carefully.
>
> At this point we believe that we have responded to all the comments and questions. Please let us know if not. For now, we will focus on editing the paper. We look forward to hearing back. Any further push-back / advice / suggestions are welcome, particularly for content that could be moved to the Appendix in favour of more valuable content.
>
> Thank you all once more for taking the time to review our work.

---

> ### Author Response · Authors · 2024-11-25
> **"Using the characterizations to select fairness parameters on a real data set"**
>
> Thank you for your comments. We have been working on a clearer discussion around $\alpha$, but it seems the empirical results are more pertinent. We are planning to repeat Speicher et al. (2018) Figures 4 and 5. More specifically, we can get $\boldsymbol{y}$, $\boldsymbol{\hat{y}}$ and  $\boldsymbol{z}$ for the Adult and COMPAS datasets, and calculate the model accuracy, index value, and between-group fairness, across thresholds, for different parameter choices. We would as you suggest compare,
>
> [$\lambda$, $(b_-,b_+)$, $\alpha]\in$  { [ accuracy, (0,2), 2 ], [ accuracy, (0.5,1.5), 0 ], [ reward rate, (close to zero, close to one), 0.5 ] }
>
> which represent choices from Speicher et al. (2018), Jin et al. (2023)  and one based on the analysis in our paper respectively. Our results show that $(b_-,b_+)=(0.01, 0.9)$ produces monotonic functions of $\mu$ for $\lambda\in[0.1,1]$. Note the reward rate is either the model acceptance rate or model rejection rate depending on whether the algorithm is assistive or punitive and the minimum benefit corresponds to the error we wish to avoid, false negatives and false positives respectively.
>
> Would this satisfy the remarks regarding empirical evidence?
>
> We welcome any suggestion of a simpler experiment if you have one in mind.

---

> ### Comment · Reviewer_cEeF · 2024-12-02
> **Response to updated draft**
>
> I thank the reviewers for putting in the time to include some empirical demonstrations of the results in their work. Unfortunately, I still feel that this submission is not ready for publication at ICLR without major revision. The primary issues are
> 1. While empirical results are now included, no details or even discussion on the results are given, making it hard to have informed conclusions from the results. Additionally, some of the plots are even missing sufficient labels to make them readable.
> 2. I still feel that my initial concerns on the numerous writing issues have not been met. To reiterate the paper is quite dense but includes sections/paragraphs that feel almost tangential or off-topic (e.g the the current version of the discussion or lines 217-220 being dedicated to a discussion on celsius vs kelvin). I would also point out to the authors the current version is not within the page limit of ICLR.
>
> With the empirical work I do think that the paper has potential, but these results still need to be worked in to the paper and the general presentation improved as well.

---

> > ### Author Response · Authors · 2024-12-02
> > **cEeF Response to updated draft**
> >
> > Thank you for your comments. What one considers to be major revisions will vary from one person to the next and some clarification here would be useful. We would hope that being over the 10 page limit by only 4 lines in itself (which could be rectified by deleting parts of the paper) should not be a major issue? We believe that the paper is not far from finished based on the changes made in response to reviewers detailed feedback over the last few weeks. Many issues have been addressed, though it remains to finesse the discussion (as we had intended) and add a conclusion (which we agree, with reviewer uB5Z, would be a valuable addition). The remaining issues as far as we can tell are on the last page of the main text and in the Appendices, we would appreciate the reviewers time in making comments and asking questions on these.
> >
> > On the numbered points:
> >
> > 1. We included the empirical results in the Appendix and did not have time to add captions and a discussion of them. We prioritized including the results over explaining them, hoping the extended discussion period would be an opportunity to clarify their meaning, as we did [here](https://openreview.net/forum?id=6jA1R0Z1G2&noteId=9suMSYVYiN)).
> >
> > 2. For convenience we paste the paragraph from Section "2.2 Mapping Predictions to Benefits" to which the comment relates:
> >
> > "A key component of the measure, is the definition of the mapping from algorithmic prediction to benefit. Benefits are floored at zero and the mean benefit must be greater than zero. Benefits are relative, they must be defined on a *ratio scale*, as oppose to an *interval scale*, to ensure that relative comparisons of benefits are meaningful. On a ratio scale, zero represents a true minimum. On an interval scale, zero is arbitrarily chosen, nevertheless differences can be interpreted meaningfully. An example is temperature, for which Kelvin is a ratio scale; Celsius and Fahrenheit are different local interval scales. If we are interested in global solutions, we should use Kelvin."
> >
> > We would argue that the measurement of temperature is not tangential / off-topic, rather it is very much connected. One cannot use Celsius or Fahrenheit to measure temperature in the [Boltzmann distribution](https://en.wikipedia.org/wiki/Boltzmann_distribution), for exactly the same reason. Measuring a human trait or ability is a physical problem just as measuring temperature and entropy are and it provides a perspective on how to think about what benefits represent when understanding fairness. That said, hopefully, deleting or replacing a sentence or two would not constitute a major revision.
> >
> > We would be grateful if the reviewer would point to the other "sections/paragraphs that feel almost tangential or off-topic" and areas where the "general presentation" is problematic.

---

> > > ### Comment · Reviewer_cEeF · 2024-12-02
> > > **Response to reviewer cEeF**
> > >
> > > *We included the empirical results in the Appendix and did not have time to add captions and a discussion of them. We prioritized including the results over explaining them, hoping the extended discussion period would be an opportunity to clarify their meaning, as we did here).*
> > >
> > > While I understand that a rebuttal period can be intense, frankly empirical results in a paper with no discussion or details (or even proper labels or captions on the figures) are not very useful/ can't really be reviewed properly and I believe I should evaluate the current draft as it stands. I am a bit confused because my initial comment asked for the results so I believe there was ample time to include both experiments and the needed discussion on them. I would also point out that reviewer RM3G pointed this out in an initial review as well, so I am not alone in this feeling.
> > >
> > > *Many issues have been addressed, though it remains to finesse the discussion (as we had intended) and add a conclusion (which we agree, with reviewer uB5Z, would be a valuable addition).*
> > >
> > > I am again a bit confused here, I agree with this reviewer that an actual conclusion is needed, and they requested one in their initial rebuttal. Why not add one over the two weeks? There seems to be push back to implementing reviewer comments which is why it is hard for me to evaluate the paper under the assumption that the comments we are making will actually be included.
> > >
> > > *We would be grateful if the reviewer would point to the other "sections/paragraphs that feel almost tangential or off-topic" and areas where the "general presentation" is problematic.*
> > >
> > > Perhaps my wording here is a bit harsh, I will rephrase my general complaint and try to give more examples that can help with this. I think there are some good ideas in the paper but that they are very muddled by presentation choices. While presentation is a personal preference, and I understand the authors disagree, these are the general suggestions I have.
> > >
> > > 1. To reiterate, the discussion section in its current form is not helpful for the reader, and really should be replaced with a focused conclusion. I understand this is on the last page but this is an important part of a paper.
> > > 2. The writing needs more focus. I included the temperature as a simple example of this, but the paper generally feels as though it alternates between very dense sections and sections of discussion that are not crucial to the paper. I gave the temperature example, but others could include the paragraph that follows or the opening of the discussion section. I understand these are minor individually, but together they can and do drag on the readability of the work.
> > > 3. The inclusion of any figures or discussion on results on real data in the main text (or even appendix) would I think help to focus the discussion quite a bit, and would be more useful than many of the current figures which are simple curve visualizations.
> > >
> > > Together, I believe these constitute major revisions but I understand this is simply an opinion.

---

> > > > ### Author Response · Authors · 2024-12-03
> > > >
> > > > *While I understand that a rebuttal period can be intense, frankly empirical results in a paper with no discussion or details (or even proper labels or captions on the figures) are not very useful/ can't really be reviewed properly and I believe I should evaluate the current draft as it stands.*
> > > >
> > > > We can only apologise for the missing information which is shared in our [comment](https://openreview.net/forum?id=6jA1R0Z1G2&noteId=9suMSYVYiN). While the graphs are not perfect they are not incomprehensible alongside the comment. All the axes are labelled. It seems a waste to ignore the information in our comment and evaluate the draft pdf alone but that is of course your prerogative.
> > > >
> > > > *There seems to be push back to implementing reviewer comments which is why it is hard for me to evaluate the paper under the assumption that the comments we are making will actually be included.*
> > > >
> > > > We have addressed many if not most reviewer criticisms as can be seen from the long discussions above. To clarify our position, we agree that something empirical would provide a valuable demonstration, but what constitutes a good experiment is subjective. We liked the suggestion by RM3G of "small demonstrative experiments somewhere in appendix" to aid understanding. Our first thought was to assume a normal distribution of scores and artificially generate $Z$, $Y$ and $\hat{Y}$ for a range of correlations, but this would not have satisfied your specific request for "real" data. Clearly it's hard to satisfy everyone, but we did attempt to during the discussion period. We implemented an improved experiment from the original work, as promised [here](https://openreview.net/forum?id=6jA1R0Z1G2&noteId=Ya3XndilSb). We will include all the proposed results (including those for the COMPAS dataset) and lines for the oracle. In short, we will implement changes in response to all the comments, of this there should be no doubt.
> > > >
> > > > On your three points:
> > > >
> > > > 1. We agree that the conclusion is an *important part of a paper* and that ours needs work, which we intend to do.
> > > > 2. We would be delighted to make the paper more readable / enjoyable by removing parts of the *discussion that are not crucial*.
> > > > 3. We disagree on the relative value you assign to the *[simple curve visualizations](https://openreview.net/forum?id=6jA1R0Z1G2&noteId=sLnULY5xn1)* (of the index representations, which are data agnostic) versus the *[empirical results](https://openreview.net/forum?id=6jA1R0Z1G2&noteId=Ya3XndilSb)* (which show results for a specific instance of of $Y$ and $\hat{Y}$). The visualizations enable a practitioner to understand exactly what a given parameter choice actually means in terms of familiar metrics (accuracy and error rates) for any problem. The empirical results represent only a single path from one point to another across the corresponding contour plot.

---

### Official Review · Reviewer_uB5Z · 2024-10-28

**Soundness:** 3
**Presentation:** 3
**Contribution:** 2
**Rating:** 6
**Confidence:** 3

**Summary:**

The authors conduct an extensive, data agnostic analysis of "generalized entropy indices" as a fairness metric. The metric, which has implications for both group and individual fairness, is dependent on parameters. Importantly -- as the authors point out -- there is little work on understanding how these parameters should be set, and what those settings imply for balances between group and individual fairness.

**Strengths:**

The core idea of the paper is strong: while the fairness community has been prolific in creating fairness metrics, there are still gaps in understanding the behavior of those metrics in different setting. I found myself wanting the same type paper, but for other well-known fairness metrics like EO, demographic parity, FPR/FNR ratios (these are not parameterized, but they do behave differently depending on underlying conditions like group base rates or classifier accuracy).

The authors also do a good job presenting theoretical claims and justifying them.

**Weaknesses:**

- Motivation: I am not fully convinced by the motivation of the paper re: why generalized entropy indices are important. The authors support their importance by noting that the metric has been implemented in libraries created by IBM, Microsoft, and Amazon — but this argument cuts both ways: it’s also true that other well known fairness packages like Aequitas and Microsoft FairLearn do not include generalized entropy indices. Instead, a much stronger way to motivate this paper would be to present a compelling example/scenario where generalized entropy indices are an ethically fair/correct fairness metric. In general, this is a core tension in individual fairness work, but it is navigable (see https://dl.acm.org/doi/abs/10.1145/3447548.3467349).

- Discussion: I found the discussion confusing and at times rambling and hard to follow. For example, on Lines 496-500, the authors write about viewing algorithmic fairness through the lens of derivative pricing — why derivative pricing? There are also strong statements like, “as a lawmaker we want to ensure the market is indeed free…” Likely many agree with statement, but it feels out of place in a paper about algorithmic fairness. I would recommend the authors choose lenses more appropriate for fairness settings.

- Discussion: In the last paragraph of the discussion, I was hoping for guidance to practitioners on how to set $\alpha$, but it fell short for me — perhaps because of the lack of context. For example, throughout the paper and discussion a “decision maker” is discussed (e.g. Line 533): who is this decision maker, what is the decision being made, what are their values, and what is the setting? If the authors grounded this discussion in a real-world fairness example, it would greatly strengthen the discussion.

- The paper ends abruptly. I would recommend the authors add a conclusion.

Overall, I am open to increasing my score if the authors can ground the paper’s motivation, findings and discussion in a compelling real-world example where generalized entropy indices are the correct choice of metric.

**Questions:**

There appears to be several abuses of notation (as well as errors) that made reading confusing... can the authors please clarify on the following:

- Each individual $i$ has a benefit $b_i$ (line 122), $\mathbf{b}$ is a benefit array (line 123), but then benefits are described with two subscripts $b_{ij}$. , $b$ is in the set of $b_{-}, b_{+}, 1$ (line 238 and 243). Is $b$ is being used to describe both individual benefit (elements of the benefit array), as well as the benefit matrix?

- On line 297 $b$ is then called as a function $b(p,y)$. Perhaps using a different variable for the benefit function, the individual benefit, and the benefit matrix would add clarity?

- Line 313 $b$ is equal to the set of $b_{-}, b_{+}, 1$. I'm guessing this is just a typo where = was used instead of $\in$?

- The assignment $b_{ij} = ((1,b_{-}), (b_{+},1))$ (line 368) makes sense in context, but is confusing against the way $b_i$ was previously defined (line 122). Does it make sense to again use a different variable to denote the benefit matrix?

- The notation used changes in the discussion. For example, $\hat{Y}$ is defined as the predicted target (line 239), but then $\hat{Y}$ is a re-defined as a model (line 490). Throughout the paper $\lambda$ is the risk-free reward rate (line 342), but then $\lambda$ is re-defined as the model accuracy (line 515). Ultimately, these changes are understandable in context, but perhaps the authors could make the paper stronger/clearer if notation was unified throughout.

---

> ### Author Response · Authors · 2024-11-15
> **uB5Z weaknesses**
>
> Hopefully, we were able to convince you of the value of GEI in our earlier comment re. motivation. We disagree that the argument cuts both ways. Our point was that the metric is well known and available to use (for those inclined) at some of the most influential companies of our time. In some cases the hardcoded parameters indicate that the model is likely being misused. Not all libraries will have implemented it, but this does not make it less important. In fact, we would argue that the lack of clarity around parameter choice is a good reason not to make the metric available.
>
> Thank you for sharing the paper. A quick look shows that the ideas shared in it are quite different to ours. We are interested in a way of mitigating between-group bias without knowing or referencing sensitive features at all. While randomness in a top-$k$ movie recommender system is a totally viable solution, in employment opportunity distribution, it is a much harder sell. Why? Because for recommender systems, the utility function for the decision maker and user are much closer than they are for employment tests. Introducing randomness in predictions is simply too far from current practices. A much easier sell (for both decision makers and regulators) is accepting that our target $Y$ is off-centre and our pricing method requires correcting - using a different but justified value of $\alpha$.
>
> We agree with your comments on the discussion and intend to rectify the issues highlighted in an updated version of the paper.

---

### Official Review · Reviewer_RM3G · 2024-11-02

**Soundness:** 4
**Presentation:** 4
**Contribution:** 4
**Rating:** 6
**Confidence:** 3

**Summary:**

This paper presents an in-depth analysis of generalized entropy indices, which reveals the relationship between generalized entropy indices and the predictive accuracy of ML models. As the author claimed in the paper, it provides an explicit connection between fairness metrics and cost-sensitive learning. The paper begins with a clear description of the generalized entropy index and its variants based on different $\alpha$; while the description is not part of the contribution, it provides the reader with a great foundation to continue the reading. The metric analysis is fascinating in revealing $I(\mathbf{b}|\alpha)$ as a function of model accuracy. This paper's analysis is solely based on mathematical derivation without empirical analysis; hence, no experiments are presented. However, I think it would be interesting to see the empirical connection.

**Strengths:**

The paper is clearly written and has many details and insights. The content is dense, causing some reading difficulty. But the overall reading experience is good.

The paper reveals the connection between the fairness metric (in terms of generalized entropy indices) and the model's performance (accuracy) with explicit expression. The explicit connection might be used to unlock future research direction on improving fairness proactively during model training (under the umbrella of cost-sensitive learning).

The paper states the potential problem of misusing generalized entropy indices with wrong parameter choice, which interests me.

Overall, the paper presents many potentially interesting insights to many people working on fairness research.

**Weaknesses:**

One of the obvious weakness is that the paper lack empirical support on the analysis. While rational analysis is good, it is often hard to be linked to practical observation one may face. I think having small demonstrative experiments somewhere in appendix can help the understanding.

The content in this paper is way to dense compare to other work I reviewed. Probably a compressed paper from a journal length work? Probably moving things from appendix into main paper will help the narrative flow. I understand this is due to the paper length limitation, but it also indicate the paper is probably more suitable for a journal publication.

Notations: some notations used in the paper is not very clearly stated. E.g. $b_{i,j}= ((1,b_{-}), (b_{+}, 1))$. Line 323. I presume this is the benefit associated with confusion matrix. But it is better clearly stated.

**Questions:**

For practical fairness evaluation, it is possible that we don't have ground-truth label or model accuracy under consideration during inference time but solely focus on individual fairness. In such a case, how would this insight presented help? Can this analysis be used for offline evaluation only?

For individual fairness, I am not very convinced the within-group component described in Equation 2 reflects individual fairness. The two data points (individuals) may belong to different groups but have strong similarities and are treated differently. Is this decomposition valid for measuring individual fairness correctly?

---

> ### Author Response · Authors · 2024-11-13
> **Index components**
>
> Thank you, we fixed the error in the introduction on line 60.
>
> - The index $I(\boldsymbol{b})$ is a measure of *individual* (overall/algorithmic) unfairness.
> - The between-group component $I^G_{\beta}(\boldsymbol{b})$ a measure of *group* unfairness.
> - In this paradigm unfairness  between groups is a portion of the overall (individual) unfairness which is given by the index.

---

> ### Author Response · Authors · 2024-11-15
> **Online evaluation**
>
> Thank you for asking this interesting question. Practically the measure of fairness investigated requires a ground truth $y_i$ to compute the benefit $b_i$. A stark difference to the notion of individual fairness described by Dwork (2012) which doesn't rely on the target $y_i$ at all. However, this work shows that they are strongly related, as we believe likely all definitions of fairness are, Binns (2019). Different definitions of fairness do not conflict but rather they assume differing knowledge in their calculation. They are in some sense different approximations of fairness, but all of them are valid and the goal should be to satisfy them *all* to the extent that we can, keeping in mind where we are now. The prioritization of them is, and always will, be context dependent. It is the role of a regulator or risk manager to determine and communicate the rules, and when they matter.
>
> To respond to a comment from reviewer uB5Z; the representations presented can be used to write the index in terms of any fairness metric imaginable, ratios and differences or whatever one chooses. The difference between this measure of individual fairness and Dwork et al. (2012) is only the dimensionality of information relied on in judging similarity between individuals; i.e., $\boldsymbol{x}_i$ versus $y_i$. The core ideology is the same. We believe, like Mukherjee et al. (2020) and other researchers, that for certain problems, independence is too strong a constraint to impose. However ignoring it completely does not make sense either. A far better expression of independence is one which demands we move in the right direction (towards independence, diversity, equality, privacy, transparency, etc.). In short, progress is a better goal than equality.
>
> We never really know our true model accuracy $\hat{Y}-\tilde{Y}$. In general we can expect to overestimate it with $\hat{Y}-Y$.  Remember that the pass-rate should always be greater than the generalization error (1 - generalization accuracy). Using too low a pass rate will almost certainly lead to a decrease in diversity - something that is well understood in recommender systems as *popularity bias*. What is a reasonable assumed generalization error? Ideally, it would be less than 50% for everybody, not just those individuals we hired in the past. One could argue that having reasonable estimate of one's generalization error and mean benefit, minimum benefit, (and choice of $\alpha$?) etc., is a reasonable ask for material people pricing models.
>
> To solve problems of fairness (manage people pricing risk), we need to use more diverse information than the decision maker. We need to compare our production model with other plausible models (including more interpretable models as a means of sanity checking). We need to measure risk (the distance between production and monitoring models) online, or periodically offline for expensive risk monitoring models. A risk manager could, in theory, train the same (production) model using this alternate model of utility and use the resulting model as an online comparative to the production model. The production system should not need sensitive features, but a diligent hiring risk manager might want to use several different human valuation models in addition to the production model as a means of risk monitoring, mitigation and reporting. They might also want to understand how the (model) valuation changes in response to changes in model parameters. Such approaches are common (and in some cases regulatory required) practices in financial institutions. While bumping all the parameters in a DNN might not be computationally feasible, bumping the final utility measure could be, providing a practical approach for risk reporting.

---

> > ### Author Response · Authors · 2024-11-17
> > **Insights**
> >
> > There are definitely clues to be found, in trying to satisfy multiple group fairness constraints as to what the problem (between accuracy and fairness) is, that lead directly to individual fairness Dwork et al. (2012) and beyond. From fairmlbook.org, we know that introducing a third possible outcome (increasing the size of our outcome space from binary to ternary), makes satisfying *independence* and *separation* possible. This tells us that we need to increase the dimensionality of our output in order to satisfy more constraints and that is what we do in our paper and with cost sensitive learning. Binary benefits allow us only to maximise for accuracy. Introducting a third possible benefit $b\in$ {$b_-,b_+,1$} allows us to account for differing error costs also. The connection with differential privacy can be seen too, in the problem with choosing a zero benefit discussed in section 2.2. Note that $f_{\alpha}(x)\rightarrow\infty$ as $x\rightarrow0$ for $\alpha\leq 0$. A zero benefit amounts to no information being exchanged - extracting information without a *user's* knowledge makes the value that can be extracted from an individual limitless under these risk models.

---

> > > ### Comment · Reviewer_RM3G · 2024-11-25
> > >
> > > Thanks for preparing your rebuttal. But I think both of my questions were not answered, leaving me concerned about the papers' core contribution. Let me recap my questions here:
> > >
> > > 1. For individual fairness, I am not very convinced the within-group component described in Equation 2 reflects individual fairness. The two data points (individuals) may belong to different groups but have strong similarities and are treated differently. Is this decomposition valid for measuring individual fairness correctly? **Is it a new definition of individual fairness? How does it align with other definitions? Why is it different from the definition in previous research?**
> > >
> > > 2. For practical fairness evaluation, it is possible that we don't have ground-truth label or model accuracy under consideration during inference time but solely focus on individual fairness. In such a case, how would this insight presented help? Can this analysis be used for offline evaluation only? **If the method is indeed as easy to apply, why don't the authors provide a simple demonstration? Any difficulty here that is not described**
> > >
> > > I realized multiple reviewers (uB5Z, cEeF and me) had similar questions about the practical concerns. While authors take time on rebuttal with long, plain feedback, a more straightforward demonstration will be more convincing and will make this paper strong.

---

> ### Author Response · Authors · 2024-11-25
> **RM3G questions**
>
> Thank you RM3G for following up. We believe we responded to both your questions in two separate comments, titled [Index Components](https://openreview.net/forum?id=6jA1R0Z1G2&noteId=3ocADkZzPq) and [Online evaluation](https://openreview.net/forum?id=6jA1R0Z1G2&noteId=3ocADkZzPq) posted on the 12th and 15th respectively. We also proposed a solution for the missing *empirical* evidence in a shared post to all reviewers titled [Empirical evidence](https://openreview.net/forum?id=6jA1R0Z1G2&noteId=sLnULY5xn1) on the 15th. On the 19th, reviewer cEeF responded stating that these were not sufficient. We post brief responses here, more detailed discussions can be found above.
>
> 1. Individual fairness is represented by the index (sum of both components) and not the within-group component. Our definition is the same as the original paper Speicher et al. (2018), we corrected a typo in the introduction.
>
> 2. If the ground truth $y_i$ is not known we cannot calculate the benefit which is a function of both $\hat{y}_i$ and $y_i$. What does it mean to "solely focus on individual fairness"?
>
> We would like to provide a demonstration but are not clear on what results would satisfy the reviewers. Our question is what results would convince you if not those proposed in our comment entitled [Empirical evidence](https://openreview.net/forum?id=6jA1R0Z1G2&noteId=sLnULY5xn1)? We had originally supposed that it was only necessary to show that the index is monotonic in $\mu$ for the parameters selected based on our analysis and provide results for a range of parameter choices, comparing our choice with that of previous authors described in Table 1, but it seems something more is required? We would be grateful if you could advise more specifically on the results / demonstration you would like to see.
>
> We have made a suggestion in the response to reviewer cEeF below entitled "Using the characterisations to select fairness parameters on a real data set".

---

### Official Review · Reviewer_XmSq · 2024-11-04

**Soundness:** 3
**Presentation:** 3
**Contribution:** 3
**Rating:** 6
**Confidence:** 4

**Summary:**

The paper builds on a previous result from Speicher et al. which provides a unified approach to quantifying unfairness at both the individual and group level. In the previous paper, the idea is to use inequality indices (from economics and social welfaire) to measure unfairness. The present paper draws connections between the inequality indices to empirical risk and cost sensitive learning. Most notably, the paper is arguing that previous work chooses arbitrary parameters for experiments, therefore this paper theoretically derives the range of index parameters and connect this to fairness guarantees. They reinterpret the original results in  Speicher et al. claiming to show that the previous empirical results do not necessarily relate to the group and individual fairness tradeoff but more generally to the trade-off between fairness and accuracy.

**Strengths:**

The paper focuses on an interesting topic which is diving deeper into a theoretical investigation of inequality indices and why the adoption and usage of the indices appears to resolve the individual / group fairness conflict. The paper provides a compelling argument for the fact that the indices relate more to the accuracy fairness trade-off.

**Weaknesses:**

The paper could be greatly improved for exposition and organizational clarity throughout.

There are also previous work showing problems with the accuracy fairness tradeoff perspective, and the paper does not seem to engage with this literature in making the argument. Why?

**Questions:**

See above

---

> ### Author Response · Authors · 2024-11-15
> **XmSq Weaknesses**
>
> To respond to your question, this work was driven by the desire to understand trade-offs analytically, in the hope of finding efficient and provable results, and so we focussed on writing complete proofs and this was the bulk of the work. There are so many papers that contributed to this work that it is difficult to engage in a meaningful way with all of them in the paper. We would be happy to address any specific unintentional omissions. We will certainly add some references in the process of editing for clarity and hopefully providing a much more enlightening discussion around $\alpha$. We would gratefully be directed to papers which are worth highlighting or expanding on.

---

### Author Response · Authors · 2024-11-13
**Notation: missing explanations and typos**

Apologies for missing notation explanations and typos. We do believe the choice to overload $b$ is the right one - the confusion is caused by unintended errors/omissions in writing, rather than the overloading itself. We answer specific questions which relate to such issues below. We will tackle broader questions about the work in separate responses which will follow.

- Line 235 should have read: A benefit function can then be defined by simply assigning a non-negative benefit value, to each element of the matrix $b_{ij}=\mathrm{benefit}(\hat{y}=i,y=j)$.
- Line 243: $b\in$ {$b_-,b_+,1$} could perhaps more clearly read, $b_i,b_{ij}\in$ {$b_-,b_+,1$}?
- Line 297: We used $b(p,y)$ only in section 3.1 for brevity/readability, favoring traditional ML notation to demonstrate the connection between empirical risk and the generalized entropy index. Would $\mathrm{benefit}(p,y)$ be clearer?
- Line 313 was indeed a typo and should read: $b_i\in$ {$b_-, b_+, 1$}
- Lines 239 & 490: We believe the notation of $\hat{Y}$ is consistent across these lines. The predicted target is based on some model or algorithm, we refer to it as a model even if the output is binary.
- In general, we treat $i$, $j$, $x$, $p$ and $b$ as dummy variables, because often they are a natural choice in the local context. In the case where we have only binary predictions we use $\hat{y}$, if we have calibrated model score, we use $p$. We use capitals for multi dimensional arrays and random variables, lower case for scalars and lowercase bold typeface for one dimensional vectors. We can add this explanation to the paper if it would help.
- At the end of section 2.2 we add the following text to clarify the definition of the *risk-free reward rate*: We shall describe the proportion of individuals receiving the unit benefit as the *risk-free reward rate* and denote it as $\lambda$. We use the terminology *risk-free*, in the sense that the benefit is known in this case to be unity. In the other cases, we do not know what the rewards $b_{\pm}$ are, they may be more or less than unity. The risk-free (unit) rewards could correspond to a column, row or diagonal. In each case, $b_{\pm}$ correspond to different (remaining) elements of the benefit matrix $b_{ij}$.
- We agree that things get confusing when one must reorient their understanding of $\lambda$, $b_-$ and $b_+$. To remedy this, we have updated our theorems to be clear about their interpretations and specifically replaced the benefit subscripts $\pm$ with $TP$, $TN$, $FP$ and $FN$.

---

> ### Author Response · Authors · 2024-11-13
> **Re: Notation: missing explanations and typos**
>
> On second thought, the terminology *unit reward rate* in place of *risk-free reward rate* would be both accurate and shorter. Thank you.

---

> ### Author Response · Authors · 2024-11-16
> **Overloading $b$**
>
> Is our preference to overload $b$ acceptable? Would different accents help (hat, bar, tilde,...) or would a different letter be necessary? $u$ and $v$ are possibilities.

---

> ### Comment · Reviewer_uB5Z · 2024-11-18
>
> I can't speak for other reviewers, but I personally found the overloading of $b$ confusing. I think being clear about defining a benefit function called $\text{benefit}$ (or using some other letter/notation, i.e. $f_b$, $f_{\text{ben}}$) would be helpful.

---

> ### Author Response · Authors · 2024-11-19
> **Overloading $b$**
>
> We did not suggest something with sub or superscripts, because these are already being used in many cases. E.g., $f_{\alpha}$ is being used for the GEI integrand, so the suggestions appear to instead overload $f_i$, note that in the Appendix we use $f_0$ and $f_1$ when considering the special cases $\alpha\in$ {$0, 1$}.
>
> Essentially we are using *tensor notation* which is better known in Applied Maths and Physics circles (https://www.cora.nwra.com/~lund/mcen5021/tensors). The subscripts are dummy variables / indices, $i$, $j$, $k$, $l$, $m$, $n$ are common choices for subscripts. Would for example replacing $b_{ij}$ with one of the following, in order of preference, suffice?
>
> 1. $b_{jk}$
> 2. $\hat{b}_{jk}$
> 3. $B_{jk}$
> 4. $u_{jk}$

---

### Author Response · Authors · 2024-11-14
**Motivation and examples**

Thank you for your feedback on our paper. With your help, we hope that we can greatly improve the clarity of it. We agree that the paper is dense (as guessed by reviewer RM3G, the paper is a condensed version of a longer work), but our hope is that it can be improved without exceeding the 10 page limit. There is clearly value in sharing a shorter exposition and we will work on this over the coming weeks. While this may not be obvious from the version of the paper which was submitted, we believe that this work could be of significant value to the community and broader society. This can be conveyed earlier in the paper to remedy feedback from reviewers uB5Z and cEeF. In particular, we have rewritten the derivatives pricing analogy, thinking instead in higher level terms of *stakeholders*. In section 2.2 we will describe these as benefit *providers*, *benefit recipients* and the *regulator*. The *decision maker* and algorithm *subject* could be the either the recipient or provider of benefits depending on the application and the level of relevance assumed in relying on test results. Hopefully the terminology is self explanatory and adds clarity to the discussion.

Although a *foreign* example, pricing risk is an important one, because there is precedent in both regulation of material (high-risk) valuation models, and best practices established in the form of regulatory required model governance by an independent risk function, public reporting requirements, whistleblower protections and more. We believe there are strong parallels between financial modelling and human rating systems. The latter should be subject to (risk appropriate) legislation, just as life insurance policies, and other derivatives, at large financial institutions are. An important question then is how to understand human rating risk so we can judge the materiality of a model. We argue that when rating a human, the benefit currency and interest rate on the transaction between decision maker and subject might not be easily described, but they exist and are implicit in the loss function choice.

We argue that generalised entropy indices (GEI) present a valuable (regulatable) family of functions (the *complete* set of subgroup decomposable functions according to Shorrocks (1980)) which warrant much closer inspection, before moving on to other welfare functions Heidari et al. (2018). We aim to prove that they parametrically extend notions of risk, in a principled and *continuous* way that allows us to manage the multiple requirements of model accuracy, fairness (differing error costs) and between-group fairness (by choice of the generalization parameter $\alpha$) in offline learning. We believe that GEI provide a parametric language ($b_{ij}$ and $\alpha$) suited to algorithmic governance at a high level. They can be computed with very little information, $(\boldsymbol{\hat{y}},\boldsymbol{y})$ or better still $(\boldsymbol{p},\boldsymbol{y})$. Such a model can be used to limit the feasible models of utility in a rational way, simply by choosing parameters reasonably and capping the index accordingly. The efficiency saving which results from using a well reasoned choice of parameters would be O($n$), since it would eliminate the need to iterate over the training data to determine the cap/threshold, which is derived analytically.

A good word reviewer cEeF used was interpretability. This is what we are trying to do with the calculation of expected cost or risk. Making the parameters interpretable so a regulator or risk manager would feel comfortable limiting their choice and interpreting the results of the calculation. As a regulator, if we have a reasonable model of algorithmic utility, we can use that to estimate how much value is being extracted with the algorithm by the decision maker at both the group and individual levels. We know that the decision maker will likely calibrate their model assuming that the cost of rejecting worthy candidates is zero. As a regulator we can make a different (fairer) assumption based on the application, and use these results to identify, challenge and mitigate algorithmic risk in employment, education and potentially beyond.

More comments will follow on remaining issues and ultimately an updated paper. Thanks again.

---

> ### Comment · Reviewer_uB5Z · 2024-11-18
>
> This expounded motivation is getting closer to convincing me of the value of GEI. But what I'm still wondering is, can you come up with a simple, clean real-world example?
>
> For example, if I wanted to justify why "False Negative Rate Disparity" is valuable fairness metric, I might motivate it with the following example: consider a setting where an algorithm is being used to predict if a high school student will fail math class, so that they can be placed into an efficacious tutoring program. In this case, False Negatives represent students who would have failed math class, but the model did not identify them to qualify for special tutoring. If there was a disparity on False Negatives based on a protected attribute, that would imply one group is unfairly missing out on access to the tutoring program.
>
> See here for more discussion on this type of motivation: https://www.datasciencepublicpolicy.org/our-work/tools-guides/aequitas/

---

> ### Author Response · Authors · 2024-11-18
> **Examples and Stakeholders, Part I**
>
> Many thanks uB5Z for sharing this link and providing a concrete example. Below are excerpts we would add to the specified sections. We can provide two examples. In both examples we are trying to avoid false positives. Note that Theorem 3.5 is new but completes the error rate distribution analysis. We add some headers related tothe examples in section 3 also.
>
> **1 Introduction**
>
> In this paper we revisit the metric proposed by Speicher et al. (2018) and mathematically prove its value in the fair measurement and regulation of risk. In order to do this we use two hypothetical examples which constitute different applications of a _sociotechnical system_ Barocas (2019). In the first, the algorithm is _punitive_, it is used to allocate harm, by determining whether or not to incarcerate individuals on trial. In the second,  the algorithm is _assistive_ (or _preventative_ Saleiro et al. (2019)), it is used to distribute employment opportunities. With these examples in mind, we consider the question of how an unfairness index _should_ behave, knowing that a cap on the index can be efficiently integrated into any convex optimization, pre-training Heidari et al. (2018). We take an intentionally data agnostic, rational as opposed to empirical Church (2011), approach to understanding the index.  Instead we focus on the abstraction of risk, represented by generalized entropy indices, and its relationship with better known performance metrics for different index parameter choices.
>
> The proposed index measure in the original paper increases the parametric representation of risk by one parameter $\alpha$. One must define a mapping from predictions to benefits (as usual when calculating risk), and specify the generalization parameter $\alpha$.
>
> **2.2 Mapping predictions to benefits**
>
> It's easiest to reason about the matrix from the perspective of one *stakeholder* at a time. We shall assume stakeholders include three broad parties. These are, the *benefit providers*, *benefit recipients* and the *regulator*. The *decision maker* and *subject* could be the either the recipient or provider of benefits. Neither benefit provider nor recipient can see beyond the decision, under one of the two outcomes. For the employer, the cost is the same regardless of whether the chosen candidate was worthy (by anyone's definition). Similarly, the cost of incarcerating a person is the same, regardless of how much the defendant earned when they were free. From any one perspective, two of the four cashflows are the same Elkan (2001). Thus, we can reduce the complexity of the analysis, by assuming that two of the four possible outcomes $\\hat{y},y\\in\\{0,1\\}$ are of unit benefit. More specifically, we will assume a ternary  model of benefits, where the benefit associated with an outcome could be one of three values, $b_{ij}\\in$ {$b_-, b_+, 1$} where $b_-<b_+$. One final constraint is that of *convexity*, for which the benefit must be monotonic in $\\hat{y}$ Heidari (2018).
>
> In this paper, we shall play the role of regulator. The decision maker exerts power and influence through deployment of their model at scale. They are, in some sense, the navigators and the stakeholders are (in most cases involuntary) passengers. As regulator, we must consider all perspectives. We accept the decision makers right to navigate (optimize), within reason or *risk appetite*. We must take, longer term view to protect everyone (including foreseeable future stakeholders) and avert disaster by constraining the direction of travel. The regulator must decide the relative importance of precision $\\mathbb{P}(Y=1|\hat{Y}=1)$ versus recall $\\mathbb{P}(\\hat{Y}=1|Y=1)$ based on the *mission*, *context* and *law*.  We can assume an unregulated decision maker would almost certainly be greedy. As the regulator, we can impose the minimum legal benefit. In some sense, every decision can be viewed as a *transaction* or *bet*; an investment (or divestment) in an *entity*, which in the future, might yield a return, or prevent a loss. The model score provides an indication of the *present value* of the subject, based on incomplete and potentially erroneous information about them. As a regulator we can preclude predatory pricing models, based on our own definition of utility, ultimately setting risk appropriate bounds on the decision space for a given application.

---

> > ### Comment · Reviewer_uB5Z · 2024-11-22
> >
> > I feel this was sufficient motivation and like the stakeholder-based analysis of benefits. I updated my score.

---

> ### Author Response · Authors · 2024-11-19
> **Minimum legal benefit**
>
> In law we already employ the concept of a *minimum legal benefit* which guarantees a reasonable minimum information exchange from decision makers. In many countries and some US states such as California, there is a requirement that the salary bands are stated in all job postings. An an entirely reasonable piece of information that candidates should have, to enable them to filter job postings. Similarly, when providing loans, some jurisdictions require a *reason* to be provided to the applicant, when a loan is rejected. The question is only how to communicate the value or currency. The minimum benefit increases with transparency - it saves people time and provides the opportunity to rectify erroneous information about them. These provide examples of policies which decision makers can implement to raise the minimum benefit in their benefit matrix.

---

> ### Author Response · Authors · 2024-11-19
> **Examples and Stakeholders, Part II**
>
> **3.2.1 Avoiding harm when algorithms are punitive**
>
> In this example, the decision maker incarcerates high risk subjects. As regulator, we wish to ensure they are not unfairly incarcerating individuals (avoid false positives). Thus, benefits should be decreasing in $\\hat{y}$, thus, $\\lambda=\\mathbb{P}(\hat{Y}=0)$.
>
> **Theorem 3.4** (Index as a function of the error distribution for $\\lambda=\\mathbb{P}(\hat{Y}=0)$ and $(b_-, b_+)=(b_{FP}, b_{TP})$)
>
> For the benefit function $b_{ij}=((1, 1),(b_{FP}, b_{TP}))$, where $b_{FP}<b_{TP}\\in(0,1)$, the index $I(\\boldsymbol{b};\\alpha)$ can be written as a function of the false negative ($FNR$) and positive ($FPR$) rates, $I(\\boldsymbol{b};\\alpha) = \\left[ p (1-FNR) f_{\\alpha}(b_{TP}) + q FPR f_{\\alpha}(b_{FP}) - f_{\\alpha}(\\mu)\\right] / \\mu^{\\alpha}$ where $\\mu = 1 - (1-b_{TP}) p (1-FNR) - (1-b_{FP})qFPR$, $p=\\mathbb{P}(Y=1)$ and $q=1-p$.
>
> **3.2.2 Avoiding harm when algorithms are assistive**
>
> In this example, the decision maker hires high scoring subjects. As regulator, we wish to ensure they are not unfairly rejecting suitable candidates (avoid false negatives). Thus, benefits should be increasing in $\\hat{y}$ thus $\\lambda=\\mathbb{P}(\hat{Y}=1)$.
>
> **Theorem 3.5** (Index as a function of the error distribution for $\\lambda=\\mathbb{P}(\hat{Y}=1)$ and $(b_-, b_+)=(b_{FN}, b_{TN})$)
>
> For the benefit function $b_{ij}=((b_{TN}, b_{FN}),(1, 1))$, where $b_{FN} < b_{TN}\\in(0,1)$, the index $I(\\boldsymbol{b};\\alpha)$ can be written as a function of the false negative ($FNR$) and positive ($FPR$) rates, $I(\\boldsymbol{b};\\alpha) = \\left[ p FNR f_{\\alpha}(b_{FN}) + q (1 - FPR)f_{\\alpha}(b_{TN}) - f_{\\alpha}(\\mu)\\right] / \\mu^{\\alpha}$ where $\\mu = 1 - (1-b_{TN})q(1-FPR) - (1-b_{FN})p FNR$, $p=\\mathbb{P}(Y=1)$ and $q=1-p$.

---

### Author Response · Authors · 2024-11-15
**Empirical evidence I**

We agree with the need for results which convince the reader that the sought after behaviour is achieved for the restricted range of parameters discussed in Theorem 3.5. We do not believe it necessary to resort to empiricism however, since with our representations, we can effectively visualise entire solution surfaces for a range of viable choices of the remaining free benefit $b$ and $\alpha$, and verify our theoretical proof. We aim to include the following in the Appendix (once again overloading our notation, this time for the index $I$):

1. Line graphs $I(\mu)$ for varying $\lambda$ which provide a side-view of the index surface.
2. Contour plots for a birds-eye-view of $I(\mu,\lambda)$.
3. Given $p=\mathbb{P}(Y=1)$, we can use contour plots to visualise $I(FNR, FPR)$.

Hopefully you agree with our preference. The first two results described above allow us to understand the results for *all* possible datasets (for a given choice of $b$, $\alpha$). We had intended to include the contour plots in the original submission's Appendix, excluding them was an oversight. We can remedy this in an updated version of the paper. Our conviction to focus on rationalism over empiricism is inspired by [Church (2011)](https://journals.colorado.edu/index.php/lilt/article/view/1245).

---

> ### Author Response · Authors · 2024-11-27
> **Empirical evidence II**
>
> We have a proposal. As discussed [below](https://openreview.net/forum?id=6jA1R0Z1G2&noteId=sgswVcwGei) we shall repeat similar experiments to Speicher et al. (2018) Figures 4 and 5. However, instead of having six subgroups, we will define a binary Z. This allows us to calculate the relevant error rate differences which represent the two different notions of fairness. Namely,
>
> 1. ```Individual unfairness:```  $I(\boldsymbol{b})$ versus $FNR-FPR$.
> 2. ```Group unfairness:```       $I^{Z}_{\beta}(\boldsymbol{b})$ versus $FNR(Z=0)-FNR(Z=1)$.
>
> Any feedback on this proposal would be appreciated.

---

### Author Response · Authors · 2024-12-02
**Updated pdf**

We thought it would be helpful to provide a quick note on the major changes to the pdf that were less discussed earlier. In particular,

- We corrected and added some discussion around $\alpha$, defining a *grit factor*  and *grit rate* at the end of section 3.
- Figures 4 - 9 in the Appendices are new.
  - There are empirical results (pertaining to the Adult dataset) in Appendix D as discussed [here](https://openreview.net/forum?id=6jA1R0Z1G2&noteId=Ya3XndilSb).
  - We should have explained in the caption of Fig. 9, that the dashed lines correspond to the right axes, and the solid lines correspond to the left. Also the colours are meaningful when dashed and solid lines are together on the same plot

We welcome comments or questions on the updates.

---

### Meta-Review · Area_Chair_EsnR · 2024-12-19

**Metareview:**

There was an extensive discussion, with reviewers somewhat split. The paper introduces an interesting point of view on fairness, but like some of the more critical reviews, I felt it could be more grounded on real data examples and evaluations - in particular, this would improve clarity of motivation. I believer the ideas are well-worth exploring, and I agree with the authors that a condensed version of a somewhat dense paper can find an audience at ICLR. However, without more clarity and motivation, this may end up not being of great service to the authors. Hopefully the heavy formalism can show a better pay-off within the shorter conference format than the current revision, and the community as a whole will better benefit from a more heavily reworked version of the manuscript.

**Additional Comments On Reviewer Discussion:**

Both authors and reviewers engaged in what I saw as a productive discussion.  The updates on the 2nd of December were acknowledged in our discussion, but the lack of a more in-depth analysis of the results still caused some uneasiness of whether the paper is ready for publication.

---

### Decision · Program_Chairs · 2025-01-22

Reject